



# Improved catalog of $NO_x$ point source emissions (version 2)

Steffen Beirle[1], Christian Borger[1], Adrian Jost[1], and Thomas Wagner[1]

[1]Satellitenfernerkundung, Max-Planck-Institut für Chemie, Mainz, Germany

**Correspondence:** Steffen Beirle (steffen.beirle@mpic.de)

**Abstract.** We present an updated (v2) catalog of $NO_x$ emissions from point sources as derived from TROPOMI measurements of $NO_2$ (PAL product) combined with wind fields from ERA5. Compared to version 1 of the catalog (Beirle et al., 2021), several improvements have been introduced to the algorithm. Most importantly, several corrections are applied, accounting for the effects of plume height on satellite sensitivity, 3D topographic effects, and the chemical loss of $NO_x$, resulting in considerably higher and more accurate $NO_x$ emissions. In addition, error estimates are provided for each point source, taking into account the uncertainties of the individual retrieval steps.

The catalog v2 is based on a fully automated iterative detection algorithm of point sources worldwide. It lists 1139 locations that have been found to be significant $NO_x$ sources. The majority of these locations match to power plants listed in the global power plant database (GPPD). Other $NO_x$ point sources correspond to cement plants, metal smelters, industrial areas, or medium-sized cities.

The emissions listed in v2 of the catalog show good agreement (within 20% on average) to emissions reported by German Environment Agency (Umweltbundesamt, UBA) as well as the United States Environmental Protection Agency (EPA). The data is publically available at https://doi.org/10.26050/WDCC/No_xPointEmissionsV2 (Beirle et al., 2023).

## 1 Introduction

Nitrogen oxides ($NO_x$=NO+$NO_2$) are key pollutants in the troposphere, affecting health as well as tropospheric chemistry. Thus, accurate and up to date inventories of $NO_x$ emissions are of great sociological and scientific interest and a prerequisite for modeling $NO_x$ concentrations accurately.

Since the mid nineties, satellite instruments measuring spectra of the light backscattered by the Earth's surface and atmosphere in the UV/vis spectral range enabled the retrieval of column densities of $NO_2$ (Monks and Beirle, 2011, and references therein). The TROPOspheric Monitoring Instrument (TROPOMI) (Veefkind et al., 2012), operated by the European Space Agency (ESA), was launched onboard the Sentinel 5 Precursor (S5-P) mission in October 2017. It is operated on a sunsynchronous orbit with equator crossing around 13:45 local time. TROPOMI provides global measurements at unprecedented high spatial resolution with ground pixel size down to 3.5×5.5 $km^2$, and high signal to noise ratio. From TROPOMI spectral measurements, tropospheric vertical column densities (TVCDs), i.e. $NO_2$ concentrations integrated vertically through the troposphere, are derived and provided as operational product (van Geffen et al., 2019, 2022).





Horizontal fluxes $\boldsymbol{F}$ can be calculated as product of TVCDs $V$ with horizontal wind fields $\boldsymbol{w}$. According to the continuity equation, the divergence of the flux, i.e. the difference between downwind and upwind flux, directly yields the balance of local emissions and sinks, as demonstrated in Beirle et al. (2019). This method is particularly sensitive for point sources, where spatial gradients are large: the spatial derivative directly yields the "excess flux" added by the point source emissions, whereas the $NO_x$ "background flux" (which might still be considerably large and complex in case of regions with traffic and industrial activities) is intrinsically accounted for. Based on this divergence method, Beirle et al. (2021) compiled a global database of $NO_x$ point source emissions, which is below referred to as v1. The catalog v1 reported 451 point sources that have been demonstrated to have high localization accuracy of about 2-3 km. However, the $NO_x$ emissions listed in v1 were far lower than those reported by governmental sources; for instance, $NO_x$ emissions for US power plants were lower than numbers reported by the United States Environmental Protection Agency (EPA) by a factor of up to 8.

The project World Emission (2022), funded by ESA, works on the quantification of emissions of various species that can be measured from satellite instruments, like $CH_4$, $CO$, $NO_x$, and several others. As part of this project, we developed an update of the $NO_x$ point source catalog, which has been improved in many aspects compared to v1. In particular, the estimated emissions are far higher and thus more realistic, as demonstrated by some regional validation, due to the combined effects of reprocessed input data, corrections for air mass factor (AMF), topography and lifetime, and a modified emission quantification procedure.

This paper is structured as follow: The used datasets are described in Sect. 2. Section 3 specifies the methods, with a focus of the improvements made in v2. The $NO_x$ emission catalog is presented in Sect. 4, and validated regionally for Germany and the USA. Section 5 discusses the performance of v2, remaining issues and restrictions of the catalog, and possible future improvements, followed by conclusions (6).

## 2 Datasets

In this section, the datasets used for the construction of v2 of the $NO_x$ emission catalog are introduced. The catalog is based on $NO_2$ TVCDs from TROPOMI (Sect. 2.1) combined with meteorological wind fields from ERA5 (Sect. 2.2). An ozone climatology (Sect. 2.3) is used for the extrapolation of $NO_2$ measurements to $NO_x$. A simple check for desert-like conditions, where TROPOMI is highly sensitive for tropospheric $NO_x$ (high surface albedo, few clouds), is made based on the TROPOMI reflectivity (Sect. 2.4).

External datasets like power plants (Sect. 2.5) and cities (Sect. 2.6) are merged in the resulting point source catalog in order to provide additional information. Finally, the derived emissions are validated against regional emission databases (Sect. 2.7).

### 2.1 TROPOMI $NO_2$

The $NO_x$ point source catalog is based on $NO_2$ TVCDs from TROPOMI (van Geffen et al., 2019, 2022) for the period from May 2018 to November 2021, using the consistently reprocessed data product provided via the S5-P Products Algorithm Laboratory (PAL) (Eskes et al., 2021) based on $NO_2$ processor version v2.3.1. Main improvement of the PAL product compared to the product versions ($\leq$ v1.3) used in Beirle et al. (2021) is the change in the cloud product due to an updated FRESCO



algorithm, generally leading to higher cloud altitudes and thus lower AMFs and higher TVCDs. In addition, "for cloud-free scenes a surface albedo correction is introduced based on the observed reflectance, which also leads to a general increase in the tropospheric $NO_2$ columns over polluted scenes of order 15%" (van Geffen et al., 2022). Both changes lead to an overall increase of $NO_2$ TVCDs of about 10-40%.

## 2.2 Meteorological data

Meteorological data is taken from ERA5 reanalysis (Hersbach et al., 2020) provided by the European Centre for Medium-Range Weather Forecasts (ECMWF). ERA5 data are used with a truncation at T639, corresponding to ≈0.3° resolution.

In order to reduce data amount, we created an intermediate meteorological dataset in which the original model output, containing horizontal wind fields (needed for the calculation of horizontal fluxes) and temperature and pressure (needed for estimating the $NO_x/NO_2$ ratio), was interpolated on a regular horizontal grid with a resolution of 1°and stored in intervals of 6 hours.

## 2.3 Ozone climatology

As in v1, "ozone mixing ratios, used for the scaling of $NO_2$ to $NO_x$, were taken from the Earth System Chemistry integrated Modelling (ESCiMo) project (Jöckel et al., 2016), using the RC1SD-base-10a simulation for the years 2000-2010. The monthly mean climatology was calculated from the model fields sampled online along the OMI-Aura overpass time (which is close to the TROPOMI overpass time) using the MESSy SORBIT sub model (Jöckel et al., 2010). As the divergence is sensitive for the added $NO_x$ at the source, the relevant $NO_x/NO_2$ ratio is that close to ground. We thus took $O_3$ concentrations from the lowest model layer." (Beirle et al., 2021).

## 2.4 Surface reflectivity

TROPOMI's directionally dependent surface Lambertian-equivalent reflectivity (DLER) v1.0 (Tilstra, 2022), based on the algorithm described in Tilstra et al. (2021), is taken from https://www.temis.nl/surface/albedo/tropomi_ler.php. The minimum LER for clear conditions, averaged over all months, at 440 nm is used for identifying regions with good observation conditions (i.e., deserts).

## 2.5 Power plant database

We use the Global Power Plant Database (GPPD) (Byers et al., 2019), in order to automatically identify $NO_x$ point sources corresponding to power plants. The GPPD lists about 35,000 power plants of all kinds, including solar, nuclear, and hydro power. For our purpose, we created a subset of those power plants using coal, gas, oil, pet coke, biomass, and waste as primary fuel, and skip power plants with capacities below 100 MW.

Basically, we make use of the latest release (v1.3) of GPPD. However, this update does not include power plants that have been shut down recently, but were still active during the time period investigated in this study. For instance, the Navajo power

plant was one of the top emitters of $NO_x$ in the US in 2019, as reported by EPA and also listed in v1 of the catalog. This power plant was shut down end of 2019, and is consequently not listed in GPPD v1.3.

Thus, we extend v1.3 of the GPPD by all power plants from v1.2 which are not included in v1.3. This adds 45 power plants, which we labeled as "(v1.2)" in the combined GPPD database.

The resulting GPPD database comprises 4741 power plants, of which 2291, 1995, and 378 use gas, coal, and oil as primary fuel, respectively.

## 2.6 Cities

In order to automatically identify cities close to the detected $NO_x$ point sources, the basic version of the World Cities Database (WCD), provided on simplemaps.com/data/world-cities, is used. Only cities with more than 100,000 inhabitants are considered.

## 2.7 Emissions

For validation purpose, we compare the $NO_x$ point source emission catalog to the following regional emission databases:

- The German Environment Agency (Umweltbundesamt, UBA) provides the Pollutant Release and Transfer Register (PRTR) for Germany (PRTR Germany, 2022). The PRTR contains annual $NO_x$ emissions of all facilities, covering energy sector as well as metal, chemical, mineral, and other industries.

- The United States Environmental Protection Agency (EPA) provides an "emissions & generation resource integrated database" (eGRID), a "comprehensive source of data ... on the environmental characteristics of almost all electric power generated in the United States" (eGRID, 2022). Here we use annual $NO_x$ emissions on plant level which are available for the years 2018, 2019 and 2020. In contrast to the PRTR, the eGRID database is focusing on electric power generation; other $NO_x$ emitters, like cement plants, metal smelters or chemical industry are not covered.

## 3 Methods

In this section, a step-by-step explanation of the point source detection and quantification algorithm for v2 of the catalog is provided. A summary of the main changes with respect to v1 of the catalog is added at the end of this section (3.13) and summarized in Table 1.

## 3.1 Data selection

As in v1, TROPOMI data is restricted to qa values (quality indicator of $NO_2$ TVCDs provided in the operational products) above 0.75, as recommended in van Geffen et al. (2019), removing cloudy pixels as well as any anomalies in the TROPOMI $NO_2$ dataset. Also the selection of solar zenith angles (SZA) below 65° is the same as in v1, restricting the TROPOMI data to favorable observation conditions for tropospheric $NO_2$ from space. In particular, over mid-latitudes, measurements in winter are skipped by this selection (Fig. 1), also avoiding complications due to potential snow cover.



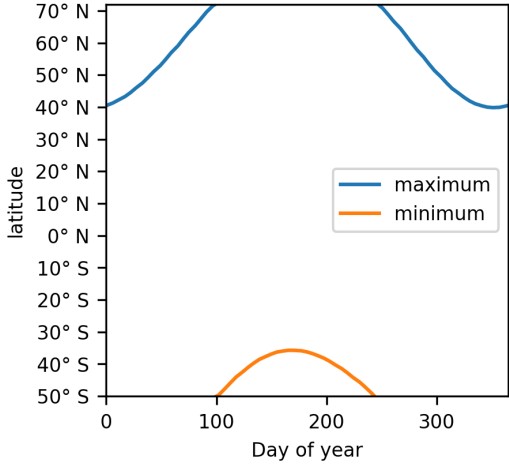

**Figure 1.** Maximum/minimum latitude of TROPOMI nadir pixels according to the SZA cutoff criterion of 65° as function of the day of year for the considered latitude range (50°S to 72°N).

.

In v2, viewing zenith angles (VZA) are restricted additionally to values below 56°, avoiding less favorable viewing conditions at the swath edges. In addition, this selection limits the maximum pixel width (across track) to 11 km. In contrast to v1, no regional preselection of potentially polluted regions was made in v2. Only high latitudes (north from 72° N or south from 50° S) are skipped ab initio.

A plume height of 500 m above ground is assumed for v2 of the catalog. This is used for the air-mass factor (AMF) correction (Sect. 3.2) as well as for the vertical interpolation of horizontal wind fields from ERA5 (Sect. 3.4). Only wind speeds above 2 m/s are considered in the further data processing. An alternative plume height of 300 m was used to quantify the uncertainty induced by the assumed a-priori plume height.

## 3.2 Air mass factor correction

The AMF and the averaging kernel (AK), reflecting the total and height dependent sensitivity of satellite measurements for atmospheric trace gases, are key concepts for the interpretation and quantification of trace gas column densities (Eskes and Boersma, 2003). The operational $NO_2$ TVCD is based on AMFs calculated for an a-priori vertical profiles of $NO_2$ taken from a global chemistry model. With the AK provided in the TROPOMI data, i.e. the ratio of height dependent (box-) AMF to the total AMF, the AMF can be adjusted to a different a-posteriori vertical profile (Eskes and Boersma, 2003).

In polluted regions, vertical profiles of $NO_2$ are generally highly complex, and a local $NO_x$ source can typically not be adequately represented by global chemistry models with comparably coarse spatial resolution. In case of $NO_x$ point sources, however, the horizontal gradient in $NO_2$ is basically sensitive to the $NO_2$ *excess* added by the point source. Any "background"

Earth System
Science
Data

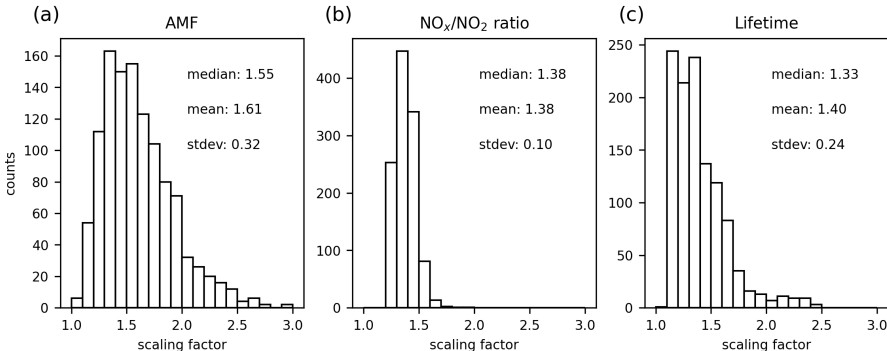

**Figure 2.** Frequency distribution of temporal mean scaling factors for (a) the AMF correction, (b) the $NO_x/NO_2$ ratio, and (c) the lifetime correction for the detected point sources.

.

$NO_2$ (which might be considerably polluted in densely populated regions) is intrinsically corrected for by the spatial derivative, i.e. the difference between $NO_2$ levels upwind and downwind of the point source.

Thus, for the quantification of point source emissions, the AMF has to be corrected with respect to the $NO_2$ *excess* added by the point source. Hence, we apply an AMF correction based on the AK at the plume altitude, assuming a default value of 500 m above ground. For the detected point sources, the AMF correction is about 1.61±0.32. Figure 2 (a) displays the respective frequency distribution for the point sources listed in the catalog v2.

Note that in v1, no AMF correction was applied, as the AK provided in the TROPOMI data used in v1 was based on a cloud height that was reported to be biased low (Compernolle et al., 2019; van Geffen et al., 2022), as discussed in Beirle et al. (2021), while no reprocessed dataset was available at that time.

### 3.3 Upscaling $NO_2$ to $NO_x$

As in v1, the TROPOMI $NO_2$ TVCD is upscaled to $NO_x$ based on photo-stationary state (PSS) according to

$$\frac{[NO_x]}{[NO_2]} = 1 + \frac{[NO]}{[NO_2]} = 1 + \frac{J}{k[O_3]}, \tag{1}$$

where

- the photolysis frequency of $NO_2$ $J$ is parameterized as $0.0167 \times \exp(-0.575/SZA)$ s$^{-1}$, as proposed by Dickerson et al. (1982), with SZA taken from TROPOMI,

- the rate constant $k$ for the reaction of $[NO]$ with $[O_3]$ is parameterized as $2.07 \times 10^{-12} \times \exp(-1400/T)$, as recommended by IUPAC (2013), with temperature $T$ (in kelvin) from ERA5, and

- $[O_3]$ is taken from a multi-year climatology modeled by ESCiMo (see Sect. 2.3).





While close to a power plant stack, PSS might not be fulfilled, it is a reasonable assumption on the spatial scales of TROPOMI pixel size (of the order of 5 km) and particularly for the 15 km radius considered for emission quantification (Sect. 3.9.1).

For the detected point sources, the $NO_x/NO_2$ ratio was found to be about $1.38\pm0.10$. Figure 2 (b) displays the respective frequency distribution. Note that these values of the $NO_x/NO_2$ ratio are not representing average values, but refer to cloud free conditions close to local noon with SZA $< 65°$.

### 3.4 Advection versus divergence

In Beirle et al. (2019, 2021), the divergence of the horizontal flux $\boldsymbol{F} = \boldsymbol{w}V$, with horizontal wind fields $\boldsymbol{w}$ and TVCD $V$, was calculated:

$$D = \nabla \cdot \boldsymbol{F} \tag{2}$$

According to product rule, this equals

$$D = \nabla \cdot (\boldsymbol{w}V) = \boldsymbol{w} \cdot \nabla V + V\nabla \cdot \boldsymbol{w} \tag{3}$$

The first term is the scalar product of (horizontal) wind vector and the spatial gradient of the TVCD; in meteorology, this is denoted as "advection" in the meaning of "the rate of change of the value of the advected property" (American Meteorological Society, 2012). Below, we use the term advection in this meaning for the quantity $A$:

$$A := \boldsymbol{w} \cdot \nabla V \tag{4}$$

The second term of Eq. 3 reflects the divergence of wind fields scaled by the TVCD. However, as we are interested in flux changes caused by local $NO_x$ emissions rather than by non vanishing divergence of the wind fields, we now directly calculate the advection according to Eq. 4, as also proposed recently by Sun (2022). I.e., in v2, the impact of non-vanishing divergence of the wind field is explicitly skipped. However, the resulting mean maps of $A$ and $D$ are very similar, as the temporal mean divergence of wind fields (at 500 m above ground) is negligibly small. Thus, the switch from "divergence" to "advection" is rather a change in terminology, but appropriately describes the retrieval steps that have actually been implemented in the processing of the catalog v2.

### 3.5 Derivative on TROPOMI grid

In Beirle et al. (2019, 2021), spatial derivatives were calculated after gridding the TROPOMI data on a regular latitude-longitude grid. In contrast, de Foy and Schauer (2022) proposed to calculate spatial derivatives directly on the native TROPOMI grid (along-track/across-track).

The main advantage of this procedure is the handling of gaps (e.g., due to cloud masking): On the TROPOMI-grid, a gap in TVCD just results in a gap in the respective gradient; in contrast, if the derivative is calculated for a temporal mean on a regular latitude-longitude grid, as in v1, gaps on individual days cause steps in the mean distribution, resulting in spikes in the spatial derivatives.



In order to calculate the advection on TROPOMI grid, the following steps are performed:

- For each TROPOMI pixel, horizontal wind fields from ERA5 are interpolated linearly to the assumed plume altitude (default: 500 m above ground), and to the observation time and latitude/longitude of the TROPOMI pixel center.

- The horizontal wind vector is transformed to TROPOMI coordinates by rotation according to the TROPOMI pixel orientation.

- The gradient of the TVCD on TROPOMI grid is calculated for each TROPOMI pixel, requiring valid TVCDs for all along-track and across-track neighbor pixels. As the TROPOMI grid becomes skewed towards the swath edges, the respective transformations of the gradient operator for skewed coordinates are applied, resulting in a scaling factor of $(1-$

$\sin(\phi))/\cos^2(\phi)$, with $\phi$ being the deviation from orthogonality (see https://en.wikipedia.org/wiki/Skew_coordinates).

- The advection is calculated as scalar product of the wind vector and the gradient of the TVCD, both defined on TROPOMI grid. The resulting scalar $A$ is independent from the coordinate system.

### 3.6 Topographic correction

In v1, systematic artifacts of the divergence map were reported over mountains with high tropospheric TVCDs, in particular
over parts of China, which hinders the identification and quantification of $NO_x$ point sources. These artifacts were explained by inaccurate wind fields over mountains in Beirle et al. (2021). However, a recent study by Sun (2022) shows that these patterns are rather caused by 3D transport effects which have been ignored so far in the simplified 2D divergence approach.

Sun (2022) derives a "topography-wind" term in order to correct for this effect:

$$C_{\text{topo}} := V/H_{\text{sh}} \cdot \boldsymbol{w_0} \cdot \nabla z_0 \tag{5}$$

with $NO_x$ TVCD $V$ (without AMF correction), $NO_x$ scale height $H_{\text{sh}}$, surface wind speed $\boldsymbol{w_0}$ and surface elevation $z_0$.

We include this correction term in order to account for topographic effects: $C_{\text{topo}}$ is calculated for each TROPOMI pixel based on the surface elevation and surface wind speed (10 m) provided in the PAL $NO_2$ data and assuming an a priori $NO_x$ scale height of 1 km.

The topography-corrected advection is then derived as

$$A^* := A + f \cdot C_{\text{topo}}, \tag{6}$$

where the scaling factor $f$ is derived empirically as 1.5 (corresponding to a net $NO_x$ scale height of 1/1.5 km = 667 m) in order to minimize topography effects, as shown in Appendix A. From now on, we denote the topography-corrected advection as $A^*$ in labels and equations, but still refer to it as "advection" in the text for sake of simplicity. I.e., the application of the topographic correction is implied below.





## 3.7 Gridding and averaging

For each TROPOMI orbit, the $NO_x$ advection derived on TROPOMI grid, as well as all other relevant variables like $NO_x$ and AMF scaling factors, wind speed, or topographic correction, are re-gridded on a regular lat-lon grid with $0.025°$ resolution, considering latitudes from $50°$ S to $72°$ N. Afterwards, temporal averages are calculated for daily, monthly, and annual periods as well as for the complete time series covered by the PAL $NO_2$ product (May 2018 – November 2021).

The temporal mean advection map of $A^*$ is the basis for the identification and quantification of $NO_x$ point sources. A high resolution map is provided in the supplementary material. Note that regions with less than 10% temporal coverage have been skipped. This criterion removes regions with poor statistics of the filtered data, caused by frequent cloud cover, snow and ice cover, and/or low wind speeds.

## 3.8 Point source identification

As in Beirle et al. (2021), point sources are identified in an automated iterative process in which local maxima of the advection map are successively checked for being point sources. The criteria for classifying potential point source candidates have been extended and modified, as explained in detail below.

In v2, a default radius 15 km is considered for the quantification of $NO_x$ emissions, in contrast to 22 km in v1. This reduces the skipping of point sources due to interfering sources nearby. A new quantity used during the categorization procedure is the "peak area fraction" which is just defined as the percentage of grid pixels (on $0.025°$grid) within a given radius around the candidate that have advection values above a threshold (here: 30% of the local maximum). This quantity helps to identify single spikes (with very low peak area fraction) as artifacts, as well as area sources (with large peak area fraction).

Point sources are identified by a fully automated iterative procedure, in which a "candidate" is identified and classified in each iteration step:

- The candidate location is defined by the absolute maximum of the advection map. Figure 3 (a) displays the maximum advection value as function of the iteration step.

- The following criteria are checked successively for the candidate until a classification is made:

  - If the distance between candidate and the edge of the advection map is less than 30 km, the candidate is categorized as "edge".

  - If more than 25% of the pixels within 15 km around the candidate are missing, it is categorized as "gap".

  - Systematic biases in e.g. the assumed plume height, ERA5 wind direction, or non-steady state effects can cause dipole-like patterns of enhanced positive and negative advection. In order to avoid such artifacts to be interpreted as point source, candidates with large negative advection values nearby are skipped. As these effects can affect larger areas, the search for negative values is extended over a larger distance: if negative values are found within 30 km around the candidate with an absolute value larger than 50% of the candidates maximum, it is categorized




(a)

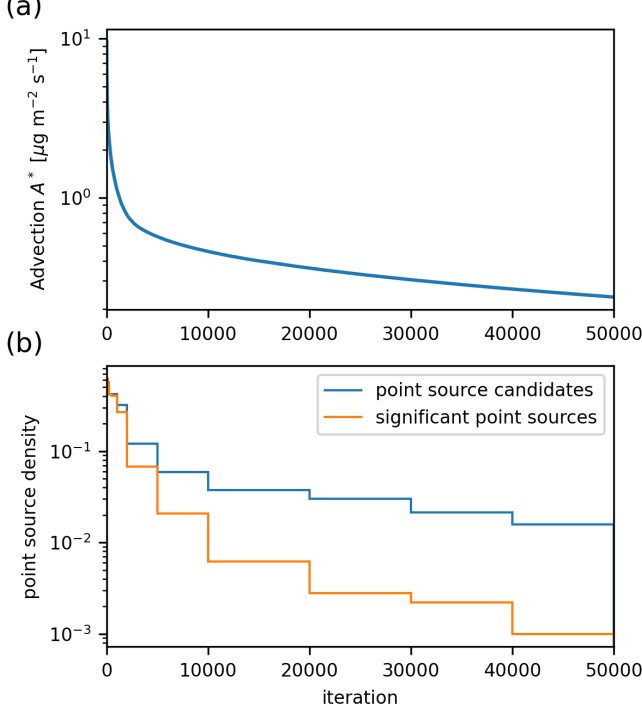

(b)

**Figure 3.** Iterative candidate classification. (a) Local maximum advection as function of the iteration step. (b) Density of point source candidates (blue) and significant point sources (orange, see Sect. 3.10) per iteration as function of the iteration step.

.

as "negative". In addition to identifying dipole-patterns, this criterion also adopts to the local noise level in the advection map and prevents the interpretation of a local maximum just caused by noise as a point source.

– If the peak area fraction within 5 km is lower than 80%, the candidate is classified as "None". This reflects spikes that do not correspond to the expected extent of the peak in advection map according to TROPOMI spatial resolution of the order of 5 km. Note that this category is very rare: only 191 out of 50,000 candidates fall into this category (Fig. 4 (b)).

– If the peak area fraction within 15 km is above 45%, indicating a spatially extended advection peak, the candidate is categorized as "area" source.

– Otherwise, the candidate is classified as point source ("ps").

– Before the next iteration step, the candidate is removed from the advection map by setting all *positive* values within 15 km (30 km in case of "negative" category) to Not a Number (NaN). Negative values are kept in the advection map such that they can still trigger the "negative" category for following candidates in the vicinity.



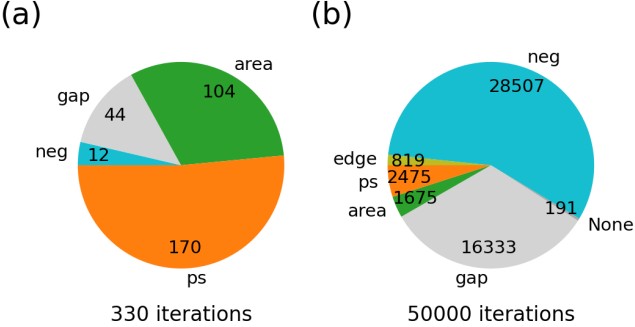

**Figure 4.** Frequency distribution of the different categories for (a) the first 330 iterations, where maximum advection is > 2 $\mu$g/m$^2$/s, and (b) for all 50,000 iterations.

.

For the catalog v2, 50,000 candidates have been processed. In the beginning, a high fraction of candidates is classified as point source (Figs. 3 (b), 4 (a)). For the first 330 iterations, where maximum advection is > 2$\mu$g/m$^2$/s, 52% of all candidates are found to be point sources, and another 32% as area source. In later iteration steps, while maximum advection decreases by almost two orders of magnitude, the majority of candidates is classified as gap (initially due to actual gaps in the input data, later due to the removal of prior candidates nearby) or negative (due to artificial dipolar patterns and due to maxima close to the local advection noise level). Within iterations 40,000-50,000, only 10 significant point sources have been found, and further iterations are not meaningful.

### 3.9 Point source quantification

For the candidates identified as point source, the respective $NO_x$ emissions are quantified by spatial integration of the advection map (Sect. 3.9.1), corrected for $NO_x$ lifetime (Sect. 3.9.2).

### 3.9.1 Spatial integration

In v1, a 2D Gaussian was fitted to the mean divergence map for the quantification of point source emissions. However, this procedure requires good statistics (i.e. long term means) and a sufficiently large spatial range (22 km radius in Beirle et al., 2021) in order to perform stable fits. Moreover, an additive background was included as fit parameter in the model function. This counteracts the paradigm of the advection (or divergence) method being sensitive to local emissions (excess $NO_x$) and that the background is already corrected for by the spatial derivative.

Thus, we have simplified the calculation of emissions by just integrating the advection map spatially 15 km around the point source location. This radius has been found to be a good compromise as it is large enough to cover the observed point source peaks in the advection map, as illustrated exemplarily for some selected point sources in Fig. 5. On the other hand, neighboring sources can still be discriminated. For instance, the Weisweiler power plant South-West of Niederaußem/Neurath (Fig. 5 (b))





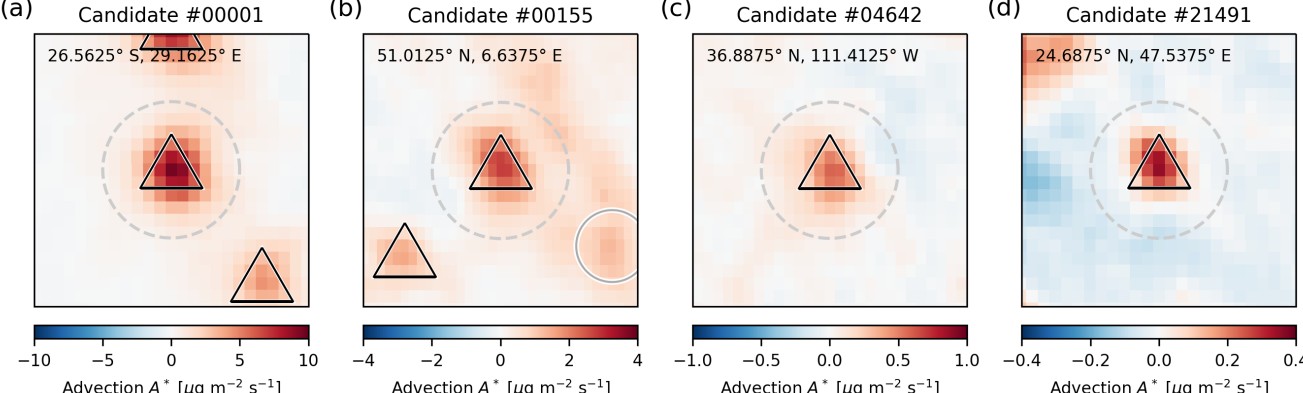

**Figure 5.** Sample maps of the temporal mean advection, corrected for topography, for (a) the first candidate classified as point source, i.e. the Secunda coal liquefier (South Africa), (b) Niederaußem/Neurath power plants (Germany; see also sect. 4.2.1), (c) the Navajo power plant (USA; see also sect. 4.2.2), and (d) the candidate with lowest derived emissions, i.e. the Al Yamama cement factory (Saudi Arabia). Results of the candidate classification are indicated by triangles for point sources and circles for area sources. The large dashed circle reflects the 15 km radius used for candidate classification procedure as well as for spatial integration. Note the different color scales.

.

is automatically detected as separate point source which was not the case in v1 of the catalog that was based on the Gaussian fit within 22 km radius. In addition, this simple and robust procedure does not rely on fit convergence and thus also works for higher spatial noise levels, i.e. for shorter temporal averages like monthly means.

We checked the impact of the simplified emission estimate procedure by applying it also to v1 of the catalog. Resulting emissions from Gaussian fit vs. spatial integration agree well with a correlation coefficient of $r=0.96$, whereby emissions from spatial integration are higher by 12 % on average.

### 3.9.2 Lifetime correction

Tropospheric $NO_x$ has a rather short lifetime of the order of some hours. Thus, the positive advection caused by a point source is opposed by the chemical loss of $NO_x$ within the downwind plume.

In Beirle et al. (2019), it was proposed to correct for the $NO_x$ loss by adding a sink term $S = V/\tau$ to $D$ by assuming a first order lifetime $\tau$. However, this approach had the disadvantage that the high-contrast maps of $D$ (or $A$) were overlaid by the spatial distribution of the mean VCD $V$ which is smeared out spatially. Consequently, the sharp contrast of $D$ (or $A$) was lost, and for integrated emissions, larger scales than a 15 km radius had to be considered. Thus, in Beirle et al. (2021), no lifetime correction was applied, as the correction was assumed to be small for strong point sources, based on a constant lifetime of 4 h. However, there are indications that the lifetime of tropospheric $NO_x$ can be significantly shorter than that. For instance, Goldberg et al. (2019) report a lifetime of only 1.5 h for the Colstrip power plant. Recently, Lange et al. (2022) systematically





investigated $NO_x$ lifetimes worldwide and found typical values of about 2 hours for low latitudes up to about 4-6 hours at higher latitudes.

In v2 of the catalog, we apply an alternative approach for correcting for the chemical loss of $NO_x$, which is based on the residence time $t_r$ of the emitted $NO_x$ within the 15 km radius:

$$t_r := \frac{15 \text{ km}}{w} \tag{7}$$

with mean wind speed $w$.

The lifetime correction has to compensate for the *integrated* loss within the residence time, which results in a scaling factor

$$c_\tau := \exp(t_r/\tau). \tag{8}$$

as explained in more detail in Appendix B.

For the calculation of $c_\tau$, we use the dependency of $\tau$ on latitude as derived by Lange et al. (2022):

$$\tau = 1.0089 \times \exp(0.0242 \times (|\text{lat}| + 9.6024), \tag{9}$$

with $\tau$ in hours and latitude in units of degree.

For the detected point sources, the resulting lifetime correction factor is about 1.40±0.24. Figure 2 (c) displays the respective frequency distribution.

### 3.9.3 Final emission estimate

Total emissions of a given point source are derived from spatial integration of $A^*$ around $r =$15 km, scaled by the lifetime correction factor:

$$E = c_\tau \cdot \int_{-r}^{r} \int_{-\sqrt{r^2-y^2}}^{\sqrt{r^2-y^2}} (A + 1.5 \times C_{\text{topo}}) \, dx \, dy = c_\tau \cdot \int_{-r}^{r} \int_{-\sqrt{r^2-y^2}}^{\sqrt{r^2-y^2}} A^* \, dx \, dy \tag{10}$$

Note that the spatial integration of the gridded advection map is realized by summing up all grid pixels within the 15 km radius.

### 3.10 Selection of significant point sources

The iterative classification algorithm yields 2475 point source candidates. For v2 of the catalog, we select significant and reliable point sources by different criteria, i.e. the detection limit, the integration error, the contribution from topographic correction, and the temporal persistence of the derived emissions:

### 3.10.1 Detection limit

In Beirle et al. (2019), the detection limit (DL) for $NO_x$ point sources was estimated to be "0.11 kg/s down to 0.03 kg/s for ideal conditions." From exemplary visible inspection, we found that these thresholds are meaningful for the updated results as well, and apply them in order to consider a point source as significant for v2 of the catalog.

"Ideal conditions" are found for cloud free scenes with high surface reflectivity, like for the Saudi-Arabian capital Riyadh (Beirle et al., 2019). We thus apply a simple albedo mask in order to decide whether a candidate faces desert-like conditions or not: For all candidates with a minimum LER above 8%, a DL of 0.03 kg/s is set, whereas for all other sites, DL is taken as 0.11 kg/s. In Fig. 8, the regions with DL of 0.03 kg/s are marked.

### 3.10.2 Integration error

For each grid pixel, temporal mean and standard deviation of all gridded quantities are calculated. This allows to calculate the standard mean error of the mean advection for each grid pixel, and thus the statistical error of the spatial integration. Point sources are considered to be significant only if the relative error of spatial integration is below 30%.

### 3.10.3 Topographic correction

The topographic correction has been found to improve the mean advection map and thus the point source emission estimate. However, the empirically derived scaling factor (Appendix A) has been selected as compromise and does not work perfectly everywhere; on contrary, some new artifacts are introduced in the advection map over mountains downwind of strong sources. In order to avoid the misinterpretation of such topographic effects as point source, we consider point sources to be significant only if the topographic correction contributes less than 50% to the integrated emissions.

### 3.10.4 Temporal persistence

For each detected point source, a time series of monthly emissions is calculated according to Eq. 10. Note that the monthly mean advection is usually too noisy in order to perform the automated point source detection; for the known point source locations derived for the full-time advection mean, however, the emissions can still be calculated on monthly basis for most cases (with higher uncertainties).

The monthly mean emissions are then checked for significance, i.e. emission values above detection limit with relative integration errors below 30%. In the final catalog, information on the number of months with significant emissions is provided. We consider this quantity as measure for temporal persistence, i.e. how persistent the point source is over time. Generally, the number of months with significant detection is the lower the weaker a point source is, as noise becomes more important. But low persistence might also indicate that a power plant was switched off during the considered period, like the Navajo power plant in the US, as shown in Sect. 4.2.2.





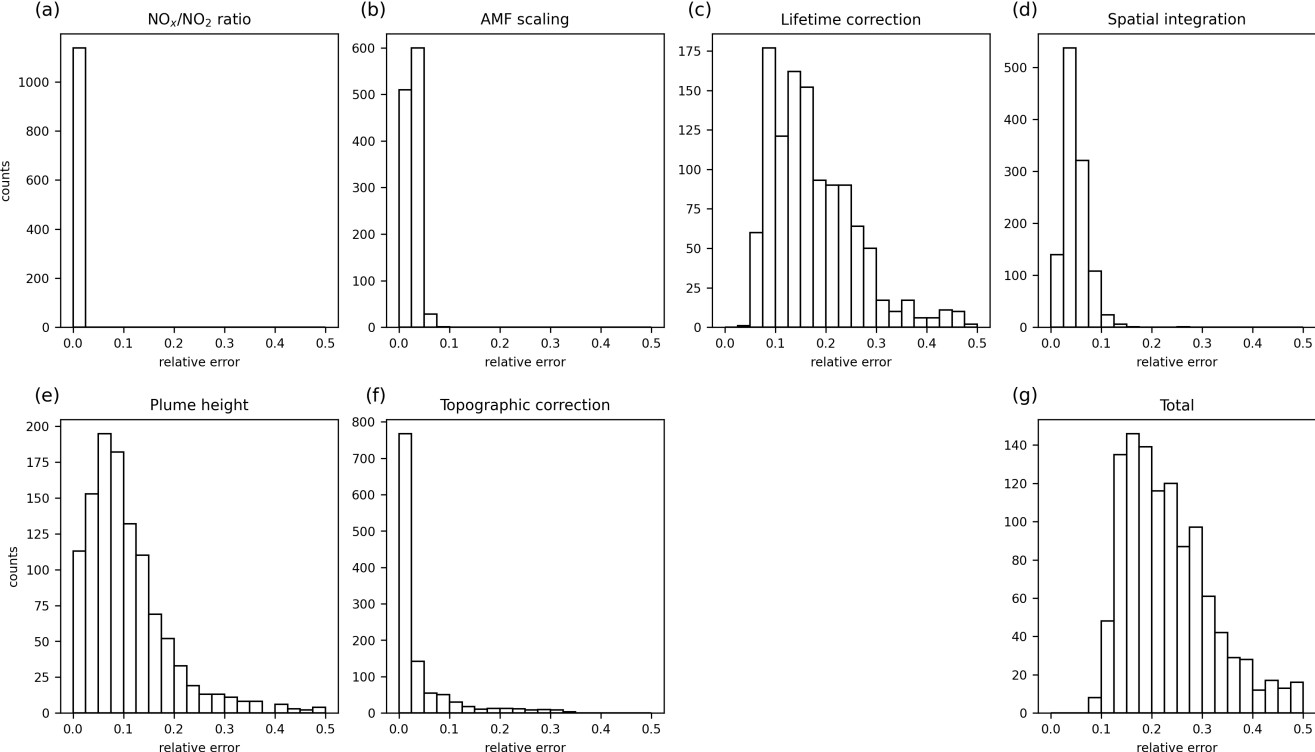

**Figure 6.** Histograms of relative errors for (a) the $NO_x$ scaling factor, (b) the AMF scaling factor, (c) the lifetime scaling factor, (d) the spatial integration, (e) the impact of a-priori plume height, (f) the topographic correction, and (g) the total error for the detected point sources.

.

Very low persistence of values down to 1, however, can also be related to exceptional events like strong biomass burning, e.g. in South America or Australia. For v2 of the catalog, we only consider point sources that show significant emissions for at least 6 months.

From the 2475 point source candidates, 1139 significant point sources remain after applying these criteria. For the catalog of $NO_x$ point sources, the remaining candidates (which are sorted by maximum advection) are re-sorted by the determined emissions (spatially integrated and lifetime corrected), and are assigned by a "rank" starting at 1.

### 3.11 Errors

The catalog v2 provides an error estimate for each derived emission. This error is calculated from the estimated uncertainties of the involved retrieval steps, as detailed below.

Note that there are further uncertainties that may cause a systematic bias of the derived emissions, but cannot be easily quantified, and are thus not included in the quantitative error estimate of the catalog. A discussion of these errors is provided in Sect. 5.3.

### 3.11.1  Scaling factors

For the scaling factors for the $NO_x/NO_2$ ratio and the AMF correction, the uncertainty is estimated from the statistical error of the temporal means for each point source. The uncertainty of the lifetime correction is calculated by error propagation applied to Eqs. 7 and 8, with the statistical error of the temporal mean of $w$, and assuming a relative uncertainty of 50% for the lifetime parameterization with latitude.

### 3.11.2  Spatial integration

The error of spatial integration is determined via the statistical error of the temporal mean for each grid pixel (see Sect. 3.10.2).

### 3.11.3  Plume height

For the plume height, an a priori value has to be assumed. The catalog v2 is based on a plume height of 500 m above ground. This height is used for two different retrieval steps:

– The application of the AMF correction, and

– The interpolation of wind fields.

In order to estimate the impact of the a priori assumption, we also performed the analysis for a plume height of 300 m, and consider the difference as uncertainty proxy. Note that in Beirle et al. (2019), similar case studies were used in order to estimate the impact of height used for wind interpolation, whereby the simultaneous impact on the AMF was ignored therein. However, explicit comparison of the AMF correction factors for plume heights of 300 m and 500 m reveals that the effect on AMF is 370  very small (about 1%). Thus, the main impact of assumed plume height is indeed that on wind fields.

### 3.11.4  Topographic correction

We apply the topographic correction with a scaling factor $f$ of 1.5 and a relative uncertainty of 33% (see App. A). This uncertainty is propagated to the corrected advection map according to Eq. 6.

### 3.11.5  Total error

Following the propagation of errors, the total error is determined from the individual contributions listed above.

Figure 6 displays histograms of the different error components and the total error. Uncertainties of scaling factors for $NO_x$ and AMF as well as spatial integration error are small (<10%). The lifetime correction has an uncertainty of about 10-20%, but can also be considerably larger for some point sources. The impact of a-priori height used for interpolation of

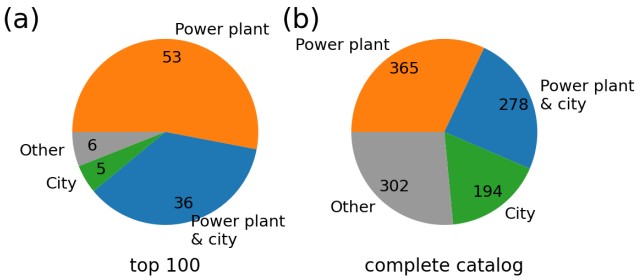

**Figure 7.** Statistic of matching power plants and cities for (a) top 100 $NO_x$ point sources and (b) the full catalog.

.

wind fields is about 10%, similar as reported in Beirle et al. (2019). The topographic correction is below 2.5% for most point
sources, but can become significant for point sources in mountain areas. Total uncertainties are typically 20-40%.

### 3.12 External information

In order to provide information about the potential origin of the detected emissions, we add spatial matches within 15 km
distance of

1. combustion power plants, as listed in the GPPD, with a capacity above 100 MW, and

2. cities, as listed in WCD, with more than 100,000 inhabitants.

Figure 7 displays the number of point sources with a match in GPPD, WCD, or both. For the top 100 point sources, a matching
power plant is found in 89 cases (in 36 cases accompanied by a city). For the complete catalog, there is a power plant nearby
still for more than half of the detected point sources, while 194 further point sources can be explained by city emissions like
traffic and/or industrial facilities. The remaining 302 point sources without a match in GPPD or WCD can correspond to cement
plants, metal smelters, or other industrial facilities outside from cities.

### 3.13 Changes of v2 with respect to Beirle et al. (2021)

Table 1 provides a comparison of the different steps for v1 and v2 of the catalog, including references to the sections where
further details are provided for v1 (Beirle et al., 2021) and v2 (this paper).

The main differences, affecting the updated catalog and in particular the reported $NO_x$ emissions, are

– the usage of the consistently reprocessed TROPOMI $NO_2$ PAL product, with higher $NO_2$ TVCDs,

– the application of an AMF correction,

– the calculation of the spatial derivative on TROPOMI pixel grid, reducing noise levels of the advection map drastically,
in particular for regions with regular cloud cover,





- the correction of topographic effects, which are considerable over mountains for high "background" pollution, like in
  parts of China or South Korea,

- the simplification of the quantification of point source emissions by spatial integration, also allowing for estimates based
  on monthly means,

- the application of an explicit correction of the $NO_x$ loss within 15 km around the point source,

- the calculation of errors for each point source.

The spatial derivative on the TROPOMI grid and the application of the topographic correction result in improved advection
maps with less artifacts, allowing for the automated detection of far more point sources (1139 compared to 451 in v1). The
higher TVCDs and the application of corrections for AMF and lifetime result in higher $NO_x$ emission estimates, resolving the
low bias that has been found for the emissions reported in v1 (Beirle et al., 2021).

## 4 Results

### 4.1 $NO_x$ point source catalog v2

Version 2 of the point source catalog can be found on https://doi.org/10.26050/WDCC/No_xPointEmissionsV2 (Beirle et al.,
2023). In addition, it is provided in the Supplement. The catalog provides latitude, longitude, $NO_x$ emissions and uncertainties
for the detected point sources. In addition, power plants from GPPD and cities from WCD are added. Also the number of
significant months is provided. Besides the basic catalog for the full period covered by the PAL $NO_2$ product, also annual
means for each year 2018-2021 are provided.

The catalog comprises 1139 point sources worldwide. Figure 8 displays an overview of the spatial distribution of detected
point sources, where power plant and city matches are color coded. In the Supplement, regional maps of the detected point
sources are provided with the corrected advection map as background image.

Table 2 shows an extract of the catalog v2. It includes the top ten emitters worldwide as well as every 100th point source
exemplarily. Figures 9 and 10 show the corresponding maps of the corrected advection.

In Appendix ??, additional tables for regional top 10 emitters are provided for various regions.

As in v1, global top 5 emitters are all located in South Africa, and all top ten emitters are related to the combustion of coal.
While the overall ranking is similar as in v1, the derived emissions are considerably higher in v2 by a factor of about 3-4 due
to the applied corrections and the new emission quantification method.

For five point sources listed in Tab. 2, no match was found in GPPD nor WCD. We checked these point sources manually
and added information about the likely $NO_x$ source, which were found to be related to coal liquefaction, cement plants, mining
activity, and/or industrial areas.

**Table 1.** Overview of algorithm steps of v2 of the catalog in comparison to v1.

| | version 1: BE21 (Beirle et al., 2021) | Sect. (BE21) | version 2 | Sect. | comment |
|---|---|---|---|---|---|
| Input data | Offline NO$_2$ product 2018-2019 | 2.1 | PAL NO$_2$ product May 2018 – Nov 2021 | 2.1 | Factor 1.1-1.4 in TVCD |
| Data selection | qa>0.75, SZA<65° | 3.1 | qa>0.75, SZA<65°, VZA<56° | 3.1 | |
| | $w > 2$ m/s | 3.5 | $w > 2$ m/s | | |
| | "regions of interest" | 3.1 | 50°S < lat < 72°N | | |
| AMF correction | None | 3.7 | According to AK at plume height | 3.2 | On average, factor of 1.61 |
| NO$_x$ to NO$_2$ ratio | PSS | 3.4 | PSS | 3.3 | |
| Quantity | Divergence | 3.5 | Advection | 3.4 | Negligible difference |
| Spatial derivative | Regular lat-lon grid | 3.5 | TROPOMI pixel grid | 3.5 | Reduced noise |
| Topographic correction | None | - | According to Sun (2022) | 3.6 | Reduced artifacts |
| Gridding | 0.025° | 3.2 | 0.025° | 3.7 | |
| Point source identification | Automated, iterative | 3.8 | Automated, iterative | 3.8 | |
| Emission estimate | Fit of 2D Gaussian | 3.8.2 | Spatial integration within 15 km | 3.9.1 | On average, factor of 1.12 |
| Lifetime correction | None | 3.6 | Based on residence time within 15 km | 3.9.2 | On average, factor of 1.40 |
| Significance criteria | Gaussian fit error | - | DL, integration error, topographic impact, persistence | 3.10 | |
| Error estimate | None | - | Explicit errors for all retrieval steps | 3.11 | |
| External information | Power plants: GPPD v1.2 | 3.9 | Power plants: GPPD v1.3 & v1.2 | 3.12 | |
| | | | Cities: WCD | 3.12 | |





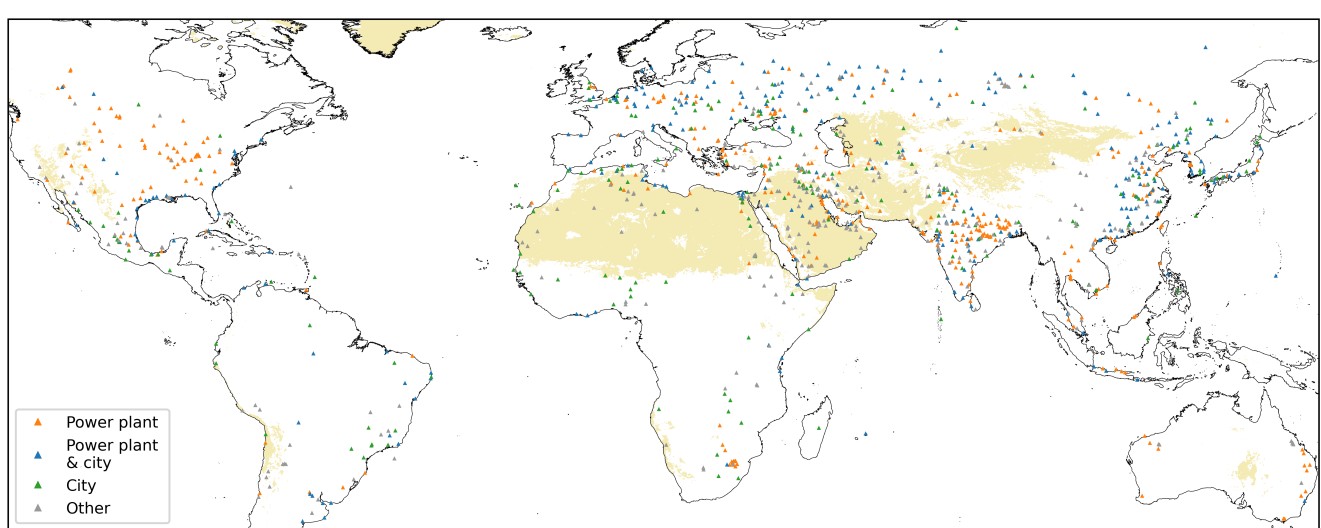

**Figure 8.** Location of point sources listed in v2 of the catalog. Matches in GPPD and/or WCD are indicated by colors as in Fig. 7. The background map highlights regions with high LER, where a detection limit of 0.03 kg/s is assumed.

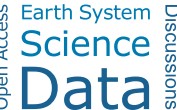

**Table 2.** Extract of the catalog v2, including rank, latitude, longitude, $NO_x$ emissions and error as well as matching power plants and cities.

| rank | lat [° N] | lon [° E] | Emissions [kg/s] | Error [kg/s] | Power plants (GPPD)[1] | Cities (WCD)[1] | Comment [2] |
|---|---|---|---|---|---|---|---|
| 1 | -26.2875 | 29.1625 | 2.76 | 0.47 | Matla; Kriel | | |
| 2 | -26.5625 | 29.1625 | 2.47 | 0.39 | | | Secunda CTL[3] |
| 3 | -23.6875 | 27.5875 | 2.47 | 0.56 | Matimba | | |
| 4 | -26.7375 | 27.9875 | 2.03 | 0.44 | Lethabo | Vereeniging | |
| 5 | -27.1125 | 29.7875 | 2.03 | 0.31 | Majuba | | |
| 6 | 22.3875 | 82.6875 | 2.01 | 0.59 | Korba | | |
| 7 | 40.6375 | 109.7375 | 1.81 | 0.57 | Baotou | Baotou | |
| 8 | 21.0125 | 107.1375 | 1.80 | 0.42 | Quang Ninh | Ha Long; Cam Pha | |
| 9 | -26.0875 | 28.9875 | 1.74 | 0.32 | Kendal | | |
| 10 | -32.4125 | 151.0125 | 1.73 | 0.30 | Bayswater; Liddell | | |
| ... | | | | | | | |
| 100 | 36.0125 | 129.3875 | 0.63 | 0.08 | Pohang | Pohang | |
| 200 | 30.8375 | 117.7875 | 0.43 | 0.10 | Tongling Wanneng | Wusong | |
| 300 | -15.0125 | -71.3875 | 0.32 | 0.11 | | | Mining facilities |
| 400 | 17.3125 | 73.2125 | 0.26 | 0.07 | Ratnagiri | | |
| 500 | 32.8875 | -79.9625 | 0.23 | 0.04 | Williams; Hagood | North Charleston | |
| 600 | 16.7375 | 43.0375 | 0.19 | 0.09 | | | Ahad Al Masarihah cement plant |
| 700 | 41.4375 | 119.5875 | 0.17 | 0.03 | | | Mining facilities; industrial area |
| 800 | 30.0625 | -94.0625 | 0.15 | 0.03 | Beaumont Refinery | Beaumont | |
| 900 | 17.9875 | -92.9375 | 0.13 | 0.04 | | Villahermosa | |
| 1000 | 47.5625 | 7.6375 | 0.12 | 0.03 | | Basel | |
| 1100 | 35.7875 | 9.8625 | 0.07 | 0.01 | | | Kairouan cement plant |

[1] Shortened for clarity.

[2] Not part of catalog v2.

[3] https://en.wikipedia.org/wiki/Secunda_CTL

**Figure 9.** Maps of $A^*$ for the top ten $NO_x$ point sources listed in the catalog (see Table 2). Markers indicate the location of point sources (triangles) and also candidates that have been discarded as area source (circles; only for advection above 0.5 $\mu g/m^2/s$). The large dashed circle reflects the 15 km radius used for candidate classification and spatial integration. Small triangles and circles show GPPD power plants and WCD cities, respectively.

.





**Figure 10.** Maps of $A^*$ for every 100th $NO_x$ point source listed in the catalog (see Table 2). Markers as in Fig. 9.

.

**Table 3.** Catalog v2 extract for point sources detected in Germany. In addition, matches with PRTR sources are added for comparison.

| rank | lat [° N] | lon [° E] | Emissions [kg/s] | Error [kg/s] | Power plants (GPPD)[1] | Cities (WCD)[1] | PRTR[1] |
|------|-----------|-----------|------------------|--------------|------------------------|-----------------|---------|
| 65 | 51.4875 | 6.7375 | 0.73 | 0.09 | Walsum | Duisburg | Steel works |
| 71 | 51.0125 | 6.6375 | 0.71 | 0.10 | Niederaussem; Neurath | | Niederaußem; Neurath |
| 168 | 53.5125 | 9.9375 | 0.49 | 0.05 | Hamburg-Moorburg | Hamburg | Hamburg-Moorburg |
| 270 | 49.5125 | 8.4375 | 0.34 | 0.05 | Mannheim | Mannheim; Ludwigshafen | GKM Mannheim; BASF chemicals |
| 368 | 50.8375 | 6.3375 | 0.28 | 0.04 | Weisweiler | | Weisweiler |
| 406 | 51.8375 | 14.4625 | 0.26 | 0.03 | Janschwalde[2] | | Jänschwalde |
| 456 | 51.4375 | 14.5625 | 0.24 | 0.04 | Boxberg | | Boxberg |
| 625 | 51.1875 | 12.3625 | 0.19 | 0.03 | Lippendorf | | Lippendorf |
| 732 | 49.4375 | 11.0625 | 0.16 | 0.03 | Franken; Sandreuth | Nuremberg; Fürth | Sandreuth |
| 746 | 53.1125 | 8.7125 | 0.16 | 0.02 | Hafen | Bremen | Hafen; Steel works |
| 775 | 50.0125 | 8.2625 | 0.16 | 0.03 | Mainz | Mainz; Wiesbaden | Mainz; Schott glass |
| 828 | 52.1625 | 10.4125 | 0.15 | 0.03 | HKW-Mitte | Braunschweig | Flat steel |
| 886 | 49.3625 | 6.7375 | 0.13 | 0.02 | Ensdorf | | Raw iron; Coking plant |

[1] Shortened for clarity.

[2] Misspelled in GPPD; should be "Jänschwalde".

## 4.2 Validation

We validate the derived $NO_x$ emissions by comparison to regional emission datasets: The PRTR for Germany (Sect. 4.2.1) and

430 the eGRID emissions for US power plants (Sect. 4.2.2).

### 4.2.1 Germany

We compare the $NO_x$ emissions of the catalog v2 to PRTR emissions reported by UBA for Germany. Each point source over Germany is merged with all PRTR emissions within 15 km. Table 3 lists an extract of the catalog for Germany, extended with the respective PRTR matches. For all point sources over Germany listed in the catalog v2, matches with GPPD as well as PRTR

sources were found.

As PRTR emissions are reported on annual basis (available for 2018-2020), we compare the annual catalog emissions to the integrated PRTR emissions within 15 km for the respective year. In Fig. 11, the catalog emissions are compared to matching PRTR emissions. A Pearson correlation coefficient of 0.81 was found between annual emissions from catalog v2 and PRTR. The ratio of mean catalog to mean PRTR emissions over all point sources and years was found to be 1.14.

For several point sources, however, interference with other emissions (in particular from traffic) have to be expected due to nearby cities, causing a high-bias of the catalog emissions. Thus, we also performed a comparison only for point sources




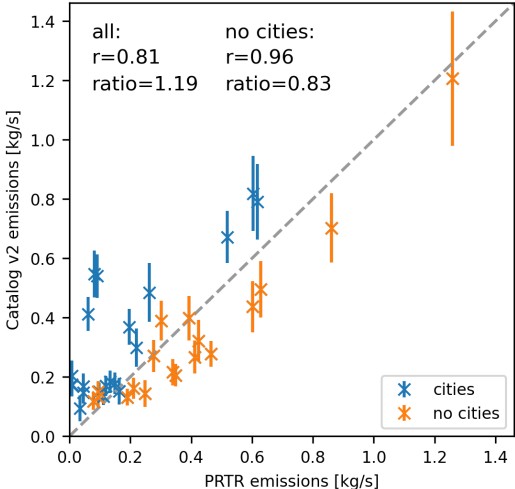

**Figure 11.** Comparison of annual mean $NO_x$ emissions from catalog v2 (y-axis) to emissions reported in PRTR, added up within 15 km radius (x-axis), for Germany. Error bars reflect the errors given in the catalog v2. Correlation coefficients r and the ratio of mean emissions (v2 versus PRTR) are provided in the figure based on all found point sources as well as for the subset excluding point sources near cities.

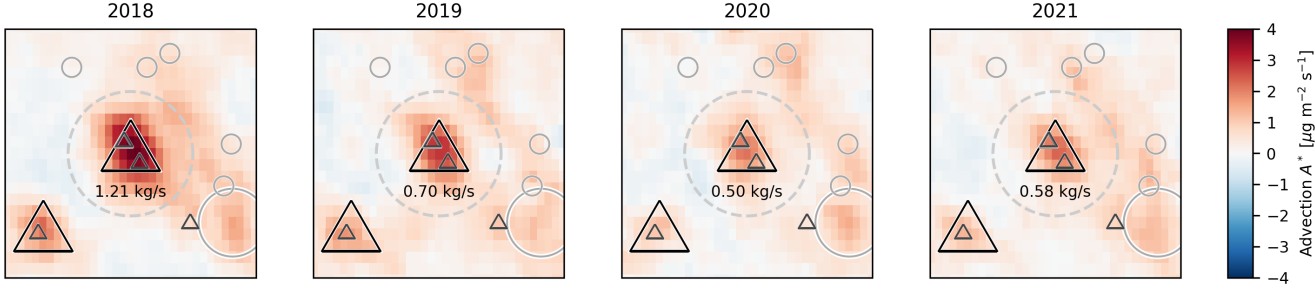

**Figure 12.** Maps of annual mean $A^*$ for catalog rank #71 (51.0125°N, 6.6375°E), corresponding to the lignite power plants Niederaußem and Neurath (see Table 3). Markers as in Fig. 9.

.

without large cities nearby. This selection of six point sources (mostly lignite power plants) increases the correlation to 0.96, while the ratio of emissions decreases to 0.83, i.e., catalog emissions are on average 17% lower than those reported in PRTR.

The highest annual emissions of 1.2 kg/s were found for the lignite power plants Niederaußem and Neurath, catalog rank
71, in the year 2018. In 2019 and 2020, these emissions decreased to 0.7 kg/s and 0.5 kg/s, respectively, in the catalog. This decrease is also reflected in the annual maps of $A^*$ (Fig. 12). A similar reduction is reported in PRTR as well.

Note that there is one additional point source listed in PRTR with an emission larger than the assumed detection limit of 0.11 kg/s which is not included in the catalog v2, i.e. the lignite power plant "Schwarze Pumpe" (51.536° N, 14.354° E). The



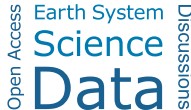

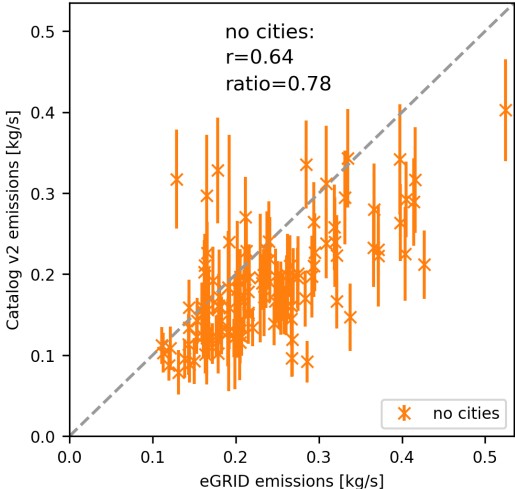

**Figure 13.** Comparison of annual mean $NO_x$ emissions from catalog v2 (y-axis) to emissions reported in eGRID, added up within 15 km radius (x-axis), for USA. Point sources close to cities are skipped, as well as eGRID values below 0.11 kg/s. Error bars reflect the errors given in v2. Correlation coefficients r and the ratio of mean emissions (v2 versus eGRID) are displayed in the figure.

location of "Schwarze Pumpe" was indeed detected as point source candidate, but was classified as "gap" due to its vicinity to
450 the "Boxberg" power plant in 18 km distance.

#### 4.2.2 USA

The eGRID dataset lists $NO_x$ emissions related to power generation, but does not cover other $NO_x$ sources from cement plants or metal/chemical/mineral industries. Thus, it has to be expected that the catalog emissions are higher than those reported by EPA whenever significant emissions from cities or industrial activities other than power generation occur within 15 km.

For a meaningful comparison between catalog v2 and eGRID, we thus focus on

- point sources that do not coincide with a large city, and

- eGRID emissions above 0.11 kg/s.

This selection keeps 41 point sources. Fig. 13 displays the corresponding comparison of annual $NO_x$ emissions between eGRID and catalog v2, resulting in a correlation coefficient of 0.64, and a ratio of mean emissions of 0.78.

In some cases, catalog emissions are larger than those reported by eGRID, probably due to interfering emissions from sources other than power plants. In few cases, the catalog emissions are considerably lower than eGRID.

The Navajo power plant was one of the top $NO_x$ emitters in 2019, but is only listed at rank #890 in the catalog v2. This is due to the shutdown of the Navajo power plant end of 2019, which also leads to Navajo being skipped from GPPD v1.3. This shutdown is well reflected in the annual emissions in v2 of the catalog (Fig. 14).





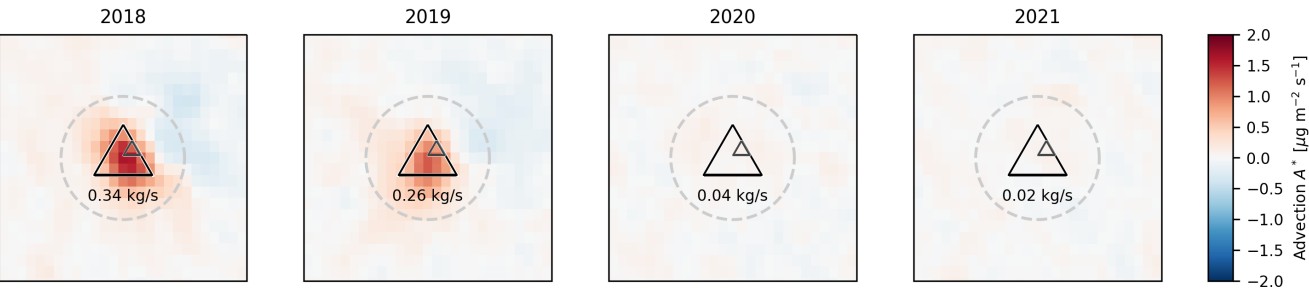

**Figure 14.** Maps of $A^*$ for catalog rank #890 (36.8875°N, 111.4125°W), corresponding to the Navajo coal power plant. The shutdown end of 2019 results in emissions close to zero in 2020 and 2021. Markers as in Fig. 9.

.

In eGRID, there are 38 further power plants listed with emissions above 0.11 kg/s that are not listed in the catalog. 36 of these power plants were identified as candidate during the iterative point source detection algorithm. In 26 cases, the candidates were indeed identified as point source, which however were not found to be significant by the strict criteria defined in Sect. 3.10, and are thus not listed in the catalog. The remaining candidates were classified as area, gap, or negative.

# 5 Discussion

## 5.1 Catalog v2

The updated $NO_x$ catalog involves several improvements compared to v1. The calculation of the derivative on the TROPOMI pixel grid avoids spikes and reduces the noise of temporal means, in particular for regions that are regularly affected by clouds. The explicit correction of topography reduces systematic artifacts over mountains. These improvements lead to higher quality map of the corrected advection which enables the automated detection of far more point sources (1139) than for v1 (451).

Due to the reprocessed TROPOMI $NO_2$ data, the applied corrections for AMF and lifetime, and the new quantification scheme of point source $NO_x$ emissions, the listed emissions are now far more realistic and agree reasonably well with reported bottom-up emissions from governmental inventories.

Thus, the catalog v2 actually provides valuable information worldwide, concerning not only the existence and location of $NO_x$ point sources, but also the respective $NO_x$ emissions. This is of particular relevance for countries where accurate emission data for point sources are not available.

## 5.2 Missing point sources

The catalog v2 cannot be expected to provide a complete list of $NO_x$ point sources worldwide for various reasons:



### 5.2.1 Data gaps

Point sources could be missing in the catalog if they are not covered by the mean advection map $A^*$. Gaps in $A^*$ can be caused
by the various selection criteria in particular for SZA, qa value (including a cloud filter) and wind speed, and the skipping of
grid pixels with less than 10% temporal coverage in the temporal mean. Note that for calculating the advection for a given
pixel on the TROPOMI grid, TVCDs must be valid for all along-track and across track neighbor pixels.

Consequently, regions with frequent cloud cover or snow and ice are missing in the temporal mean advection map, as can be
seen in the regional maps of $A*$ provided in the Supplement. In addition, there are some gaps at desert coastlines, for instance
along the Persian Gulf. This is caused by the coarse resolution of the surface albedo map used for the FRESCO cloud product.
This issue is expected to improve by the recent update of the processor version v2.4 of the TROPOMI $NO_2$ processor, as the
new TROPOMI DLER product with 0.125°resolution is used consistently for the cloud (FRESCO) and $NO_2$ retrievals, which
will improve the identification of clouded pixels over scenes with strong changes of surface albedo, in particular sand - water
transitions.

### 5.2.2 Spatial interference

Due to the integration within 15 km, point sources close to each other cannot be separated and are likely to be interpreted as
one single point source, as for instance the German lignite power plants Niederaußem and Neurath with a distance of 9 km.

For point sources within about 20 km distance from other strong point sources or large cities, spatial interference might cause
candidates to be classified as gap or be merged with the city emissions, respectively. For instance, around Riyadh (compare
Fig. 3 in Beirle et al., 2019), the automated algorithm successfully detects the power plants PP9 and PP10, both about 25 km
afar from the city center, whereas PP7 and PP8 (<20 km distance to city center) are identified as candidate, but classified as
area source due to the interference with Riyadh city emissions.

### 5.2.3 Insignificant point sources

The catalog combines the identification of point sources within a fully automated algorithm with the quantification of the
505 respective $NO_x$ emissions. In order to avoid false detection caused by artifacts or noise, stringent criteria are applied in order
to identify significant point sources. For the US, 26 of the sources listed in eGRID were correctly identified as point source
candidate, but discarded as insignificant, and are thus not included in the catalog v2.

The quantification of $NO_x$ emissions by spatial integration of the corrected advection map could be applied to these locations,
or any other known point source, as well. However, for large parts of the world, such reliable a-priori knowledge about the
510 location of point sources is not available. Thus, the catalog v2 focuses on point sources that could be identified from the mean
advection map without additional a-priori knowledge.

### 5.3 Systematic errors

The catalog v2 provides error estimates for each point source based on the estimated uncertainties for each retrieval steps. In addition, there are potential systematic errors:

### 5.3.1 Integrated emissions

The catalog integrates the corrected advection map over a 15 km radius around the identified point sources. Consequently, the reported emissions refer to all emissions within this area. In case of other sources nearby, like traffic or other industrial facilities, these sources cannot be discriminated any further, nor can the catalog indicate which fraction of the integrated emissions can actually be assigned to the point source itself without additional information about sources nearby.

### 5.3.2 Uncertainties of wind fields

As discussed in Beirle et al. (2019), an error (random as well as systematic) of the assumed wind direction causes a *systematic* underestimation of the determined flux, as only the wind component parallel to the actual wind direction matters. In Beirle et al. (2019), this effect was estimated to be about 3% for the city of Riyadh. Larger biases have to be expected over regions with low wind speeds (with higher uncertainties of wind direction) and over mountains (where modeled wind fields are generally more uncertain and spatial resolution of the meteorological model might not be sufficient).

### 5.3.3 Mountains

In addition to higher uncertainties in wind fields, also 3D effects of transport come into play as soon as the terrain has spatial gradients. Sun (2022) proposed an explicit correction term for this effect. in which the surface concentration of $NO_x$ is estimated as the ratio of the tropospheric column and an a priori $NO_x$ scale height. Here, we apply the correction term (Eq. 5) scaled by $f$=1.5, which corresponds to a net $NO_x$ scale height of 0.66 km (Appendix A).

The consideration of the topographic advection term improves the advection maps over mountains significantly. However, there are still some artifacts (both positive and negative) remaining as can be seen in Fig. A1.

For further improvements, wind fields with better spatial and/or temporal resolution should be used. In addition, the topographic advection might be applied with spatially varying $NO_x$ scale heights by using external information, e.g. from chemical transfer models.

### 5.3.4 3D effects of radiative transfer for power plant plumes

AMFs are usually calculated for a-priori trace gas profiles without the consideration of horizontal gradients and applying the independent pixel approximation. With TROPOMI, however, pixel size becomes so small that 3D effects of radiative transfer matters. As shown in Wagner et al. (2022), horizontal light paths lead to a smearing out of the satellite observations of a confined plume: TVCDs of plume pixels are generally biased low when derived with a 1D AMF, while neighboring pixels are biased high. For a plume of 1km × 1km × 1km, Wagner et al. (2022) report a low bias of up to 30% for $NO_2$ for the





TROPOMI pixel covering the plume. Note, however, that this effect is slightly dampened by the spatial integration within 15 km applied in the catalog v2.

For an accurate quantitative estimate and potential correction, further studies are required that take the specific geometry of power plant plumes into account.

### 5.3.5   Total bias

Whereas the spatial integration within 15 km may cause a high bias of the reported catalog emissions in case of interfering sources (Sect. 5.3.1), the effects described in sections 5.3.2 and particularly 5.3.4 lead to a low bias of the order of up to 30%. Thus, the observed low bias of the catalog v2 emissions of about 20% when compared to PRTR or eGRID can be understood.

Future dedicated studies of 3D radiative transfer effects for power plant plumes will allow for better quantitative corrections.

## 6   Conclusions

Based on consistently reprocessed TROPOMI $NO_2$ data for the time period May 2018 to November 2021 (PAL product), combined with wind fields from ERA5, we compiled an updated catalog (v2) of $NO_x$ emissions from point sources worldwide.

     Compared to v1 of the catalog (Beirle et al., 2021), several improvements were implemented; the most important ones are

1.  the usage of the PAL product (Eskes et al., 2021),

         2.  the calculation of spatial derivatives on TROPOMI grid (de Foy and Schauer, 2022),

         3.  the correction of 3D effects of transport over mountains (Sun, 2022),

         4.  the correction of AMF according to the AK at plume height,

         5.  the correction for chemical loss of $NO_x$, and

6.  a simplified scheme for calculating $NO_x$ emissions.

In addition, the advection, i.e. the scalar product of horizontal wind fields and the spatial gradient of $NO_x$ TVCDs is calculated rather than the divergence of the $NO_x$ flux, which has negligible effect on resulting emissions, but alters terminology.

     Steps 2 and 3 reduce noise and systematic artifacts, respectively, in the temporal mean advection maps, allowing for the automated detection of $NO_x$ point sources (1139 compared to 451 in v1). Steps 1 and 4-6 result in far higher and more realistic

$NO_x$ emissions. Comparisons to PRTR emissions for Germany and eGRID emissions for the US agree well with a remaining low bias of about 20% of the updated catalog for the detected point sources (excluding those close to cities).

     Due to step 6, also shorter time periods like annual means can be considered, and the annual emissions included in the catalog v2 do well reflect for instance the reduction of power plant emissions from Niederaußem and Neurath between 2018 and 2020, or the shutdown of the Navajo power plant end of 2019.

Future updates will focus on including wind fields with improved spatial and temporal resolution. In addition, the impact of 3D radiative transfer effects on AMFs for power plant plumes will be investigated in more detail.



## 7    Data availability

Version 2 of the $NO_x$ point source catalog can be found on https://doi.org/10.26050/WDCC/No_xPointEmissionsV2 (Beirle et al., 2023).

## Appendix A:  Topographic correction

The topographic correction $C_{topo}$ was calculated for an a priori $NO_x$ scale height of 1 km. For the calculation of topography-corrected advection $A^*$ (Eq. 6), $C_{topo}$ is scaled by a factor $f$ which is adopted empirically.

Figure A1 displays uncorrected and corrected advection maps for different values of $f$ for mountain regions with high $NO_x$ emissions, i.e. greater areas of Los Angeles, Tehran, Seoul, as well as the Chinese Shanxi province.

For the cities Los Angeles and Tehran, both exposed to high levels of $NO_x$ pollution and both close to high mountains, the topographic correction has a tremendous effect; Tehran, almost invisible in the uncorrected advection, becomes the place with highest advection worldwide (i.e., candidate #0, classified as area source) after applying the topographic correction.

For Northern China and South Korea, the uncorrected advection shows strong dipolar patterns of positive and negative advection values. These patterns caused several local maxima to be classified as "negative" in v1 of the catalog (Beirle et al., 2021). By applying the topographic correction, these patterns are suppressed, allowing for the identification of several additional point sources. However, even for $f=2$ (corresponding to a $NO_x$ scale height of 500 m), the dipolar patterns do not vanish completely. On the other hand, a high value of $f$ introduces new artifacts afar from the sources, for instance North of Los Angeles or South of Tehran. This can be understood since here the $NO_x$ scale height is larger, and the appropriate $f$ would be lower.

Without additional knowledge about the (location dependent) $NO_x$ scale height, the application of the topographic correction term is thus a compromise. For v2 of the catalog, we choose a value of $f=1.5$, corresponding to a net $NO_x$ scale height of 667 m. For the error estimate, a relative uncertainty of 33% (corresponding to $f$ in the range of 1.0-2.0, or $NO_x$ scale height in the range 500-1000 m) is assumed for the topographic correction term.



**Appendix B:  Lifetime correction**

Consider a point source with $NO_x$ emissions $E$. Following the emission plume in a Lagrangian reference frame, the amount
of $NO_x$ as function of time is proportional to $\exp(-\frac{t}{\tau})$ for a first-order lifetime $\tau$. Thus, also the chemical loss of $NO_x$ $L$ is
exponentially decreasing with time. Due to overall mass balance, $L$ can be written as

$$L(t) = E/\tau \times \exp\left(-\frac{t}{\tau}\right) \tag{B1}$$

since the integrated loss $\int_0^\infty L\,dt$ must equal the initial emissions $E$.

The integrated advection contains both, emissions and losses, within 15 km. In order to receive the emissions, the contribu-
tion of the loss term has to be quantified and corrected for. For a given wind vector $\boldsymbol{w}$, the spatial integration over 15 km radius
can be transformed into a temporal integration over the residence time

$$t_r := \frac{15\ \text{km}}{|\boldsymbol{w}|} \tag{B2}$$

Without loss of generality, the spatial integration can be expressed in a coordinate system where $x$ follows the wind direction.
Thus, the integration in across-wind direction $y$ does not contribute to the chemical loss.

The spatial integration of the advection yields

$$\int\limits_{x=-15\,\text{km}}^{15\,\text{km}} \int\limits_{y} A^* \,dy\,dx = E - \int\limits_0^{t_r} L\,dt = E + E \times \exp\left(-\frac{t_r}{\tau}\right) - E = E \times \exp\left(-\frac{t_r}{\tau}\right) \tag{B3}$$

I.e., the point source emissions can be derived by scaling the spatially integrated advection with the factor $\exp\left(\frac{t_r}{\tau}\right)$:

$$E = \int\int A^* \,dy\,dx \times \exp\left(\frac{t_r}{\tau}\right) \tag{B4}$$







**Figure A1.** Uncorrected and corrected advection maps for different scaling factors $f$ of the topographic correction for Los Angeles, Tehran, the Shanxi province, and Seoul.

.



*Author contributions.* SB designed this study, performed the analysis and wrote the paper with input from all co-authors. AJ and CB supported data processing. TW supervised the study.

*Competing interests.* The authors have no competing interests to declare.

*Acknowledgements.* This study received funding from the ESA World Emission project (https://www.world-emission.com), and the catalog of $NO_x$ point source emissions v2 is also included in the World Emission Database.



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
