# Peer review of "Improved catalog of NOx point source emissions (version 2)"

_Earth System Science Data, 2023_

## Author Comment (AC1)

**Reply to reviewer #1**

The original review is included in grey. Text changes in the revised manuscript are indicated in italic font. References to section numbers are given for both the discussion version of the manuscript (in strike-through mode) as well as the revised manuscript.

This paper presents an improved (version 2) catalog of global NOx point source emissions from TROPOMI NO$_2$ data. Point source rates are estimated on the basis of flux divergence applied to TROPOMI gradients. The paper includes detailed discussion of errors and of improvements relative to version 1, as well as evaluation with power plant data in Germany and the US. The dataset is a very useful compilation, representing a significant advance over version 1, and the error analysis is clearly presented. The Abstract and Conclusions summarize the paper well. The presentation is very good. I recommend publication after consideration by the authors of the following comments.

We thank the reviewer for his/her positive feedback. Below we reply to the specific comments point by point.

1. Figure 2: I found the presentation of AMF, NO2/NOx, and lifetime as scaling factors not helpful at all. I'm sure that these scaling factors are of interest to the authors but they are not to the reader. I recommend showing the actual quantities in Figure 2, and the scaling factors can be given as statistics in the text.

   We thank the reviewer for this critical feedback, pointing out the need for clarification of our motivation for displaying the "scaling factors", which might seem to be rather abstract quantities.
   However, we still consider these scaling factors to be an essential part of the update, as they actually explain the overall higher emission values in v2 by a factor of about 3, and quantify the impact of the different corrections applied:

   - For the AMF, the scaling factor indicates how much higher the corrected AMF (for a delta peak at 500 m agl) is over the a-priori value.
   - For the NOx/NO2 ratio, the scaling factor directly represents the "actual quantity", i.e. [NOx]/ [NO2].
   - For the lifetime correction, the scaling factor compensates for the loss of NOx within the integrated area (circle with 15 km radius). E.g., a scaling factor of 1.25 would compensate a reduction to 80% of the emitted NOx due to chemical loss within the residence time.

   In the revised version of the manuscript, we have clarified the meaning of the scaling factors as follows:

   Section  3.3:
   …
   *Hence, we apply an AMF scaling factor $c_{AMF} = AMF_{plume}/AMF_{PAL}$, where $AMF_{PAL}$ is the tropospheric AMF applied in the PAL product, and $AMF_{plume}$ is calculated from the AK based on a delta-peak profile at plume height (default 500 m). I.e., $c_{AMF}$ reflects how much higher the plume AMF is compared to the a-priori value.*

   Section  3.4:

   *… by the scaling factor $c_{NOx}$ which is calculated based on photo-stationary state …*

In addition, we now clarify at the end of section  3.4 that
*Below, we consider TVCDs of NOx (denoted as V) which are derived from the PAL NO2 TVCD multiplied by scaling with $c_{AMF}$ and $c_{NOx}$.*

Section  3.10.2:
…
*A scaling factor of e.g. $c_\tau$=1.25 thus compensates a reduction to 80% of the emitted $NO_x$ due to the chemical loss within the residence time.*

In the caption of Fig. 2, we now refer to the main text for further details about the scaling factors. In addition, instead of "scaling factor" we now use the more specific terms $c_{AMF}$, $c_{NOx}$ and $c_\tau$ as labels of the subplots.

2. Section 3.3: Inference of the NO2/NOx ratio is crude and could be a significant source of error needing to be mentioned.

We inferred the NOx/NO2 ratio based on photo-stationary state for each individual TROPOMI pixel. We consider this being far less crude than using a constant value as was done in our earlier studies (Beirle et al., 2011; 2019) as well as in several other recent publications (e.g. de Foy and Schauer, 2022; Dix et al., 2022; Goldberg et al., 2022; Sun, 2022).

Statistical fluctuations of the photolysis frequency, the rate constant, or the ozone concentration are eliminated by the consideration of the temporal mean. Any systematic deviation of J, k or [O3] from the used parameterizations/climatologies would indeed affect the resulting NOx values. However, a systematic bias of e.g. the ozone concentration of 10% would affect the NO/NO2 ratio by about 10%, but the NOx/NO2 ratio would change by only 3%. Thus we do not consider the calculated NOx/NO2 ratios to be a general major source of uncertainty for the presented results.

*J* should depend on surface reflectivity and it's a bit embarrassing that this is not recognized since it is key to the NO2 retrieval.

We agree that ignoring the surface albedo within the parameterization of *J* is a simplification that systematically affects the results for scenes with high albedo (deserts). According to Augustsson, 1981, a change of the surface albedo from typically 5% to 25 % over deserts increases *J* (and thus the NO/NO$_2$ ratio) by a factor of about 1.3. This increase corresponds to an increase of the NO$_x$/NO$_2$ ratio of about 5% (based on an a-priori NOx/NO2 ratio of about 1.2 over deserts, see Fig. 2 in Beirle et al., 2021).
In the revised version of the manuscript, we shortly discuss this systematic error at the end of a new subsection 5.3.1 discussing the PSS assumption:
*Additional systematic errors might be introduced by the parameterization of J as function of the SZA, which so far ignores the impact of the surface albedo, causing a low bias of the [NOx]/[NO2] ratio over deserts of about 5%. This parameterization might be improved in future studies.*

The ozone climatology likely does not apply to the plume where ozone would be titrated at least in the near-field (and 15 km is close to that). I see no support for the authors' claim that PSS is a reasonable assumption on the 15-km scale. Plume chemistry can be very weird. It would be good to discuss the literature for direct measurements of the NO2/NOx ratio in plumes.

We thank the reviewer for raising this issue, and agree that PSS might not always be reached within 15 km, in particular for the strongest emitters.
In the revised manuscript, we have added a new subsection to the discussion of systematic errors:

*5.3.1 Photostationary state*
*The scaling of NO2 observations to NOx is based on PSS assumption. This is typically not fulfilled directly at a strong point source due to the added emissions which take place largely in form of NO. These NO emissions are converted to NO2 during plume travel and will thus be detected by the spatial gradient not at, but downwind from the source, which would cause a smearing out of the peak in the advection map. Note, however, that the width of the advection plume of about 5 km (Beirle et al., 2021) is of the order of the TROPOMI pixel size, and we do not observe a significant downwind shift in the advection (or divergence) signal.*
*Nevertheless, we cannot rule out that PSS is not always reached completely within 15 km. Janssen et al., 1988 parameterized the deviation from PSS as function of downwind distance based on actual aircraft measurements of power plant plumes. The deviations from PSS at 15 km distance have been found to be about 2% for summer (based on α=0.25/km, see Table 4 in Janssen et al., 1988) up to 24% in spring and autumn for low background ozone concentrations (based on α=0.1/km, see Table 3 in Janssen et al., 1988), which would cause a corresponding low bias in the estimated emissions.*
*Increasing the considered radius to e.g. 20 km would reduce the possible bias of the emission estimate due to non PSS. On the other hand, this would have other negative impacts:*
*- some of the detected point sources could not be separated any more,*
*- the interference with other sources around the point source would increase, and*
*- the uncertainty of the lifetime correction, which is based on the residence time derived from the wind speed at the point source, would increase.*
*Thus, we stick to the choice of the 15 km radius in this study.*

In addition, we also list this effect in the concluding discussion in section  5.3.7, and increase the estimated overall low bias to "*up to 40%*".

3. Section 3.8: I don't understand the iterative nature of the procedure, and this is worth explaining better because it's brought up in a number of places including the abstract. As I (maybe incorrectly) see it, the procedure applies successively lower thresholds to the flux divergence thresholds. It's not really iterative except that previously identified plumes are removed from the dataset. I may not have it right, which is the point. Clarify.

We use the term "iteration" as "the repetition of a process in order to generate a … sequence of outcomes. Each repetition of the process is a single iteration, and the outcome of each iteration is then the starting point of the next iteration." (Wikipedia). The "sequence of outcomes" is the list of candidates, and the "starting point of the next iteration" is the advection map where the previous candidate has been removed. We point out the iterative nature of the plume identification in contrast to other algorithms like e.g. pattern recognition.

4. Section 3.9.2: do we understand the chemistry behind this NOx lifetime of 2 hours at low latitudes, 4-6 hours at higher latitudes? My recollection is that the mechanism for

fast NOx oxidation in power plant plumes remains a bit of a mystery considering that ozone titration would be expected. It would be good to review some of that literature.

Several studies report on a generally short NOx lifetimes of about few hours for power plant plumes based on satellite measurements (Goldberg et al., 2019, 2022; Lange et al., 2022). Similar short lifetimes of about 3 hours have been reported in Ryerson et al., 1998, based on aircraft measurements.
This is consistent with enhanced OH concentrations reported in power plant plumes (Kim et al., 2016; de Gouw et al., 2019). Thus we see no fundamental contradiction in atmospheric chemistry with respect to short NOx lifetimes.

Surely the lifetime parameterization should be a function of season as well as latitude.

We added the following paragraph to section 3.9.2 3.10.2:
*Note that the seasonal dependency of the NOx lifetime has been found to be rather weak (probably due to the focus on cloud free conditions around noon), while seasonal estimates have larger uncertainties due to reduced statistics (Lange et al., 2022). Thus, we do not consider a possible seasonal dependency of the NOx lifetime explicitly. In addition, high variability of lifetimes at different locations of similar latitude has been reported e.g. in Laughner and Cohen (2019). Thus we assume a rather large uncertainty of 50% for $\tau$ (see Section 3.12.1).*

5. Equation (10): I don't understand the double integral. Shouldn't it be a contour integral?

The point source emissions are derived from the advection map, which provides rate densities (mass per time per area), by spatial integration over the considered area of a circle with 15 km radius (yielding mass per time). Thus Eq. 10 11 does not describe a contour integral, but a double (area) integral.
We see the potential confusion by Eq. 10 11 due to the complicated integral bounds, while the corresponding procedure in the code is a simple summation of advection values of the grid pixels within 15 km, multiplied with the respective pixel area. We tried to reduce the confusion by replacing the explicit integral bounds by a more symbolic notation indicating the "circle area".

6. Section 3.11.1: I don't understand the 'statistical error' terminology in that section. Are you referring instead to variability, such as standard deviation?

The "statistical error" refers to the standard error of the temporal mean, i.e. the temporal standard deviation divided by the square root of the sample size. This is clarified in the revised manuscript.

7. Section 3.11.3: a plume height of 300 m vs. 500 m is not enough to characterize the uncertainty in the AMF. In early afternoon when TROPOMI observes, vertical mixing up to the PBL depth (typically 2 km) can take place within 1 hour. That may give you an AMF error of more like 10%.

Within the error formalism, the statistical uncertainty of the AMF is estimated from the temporal variability of the AMF scaling factor and is found to be of the order of up to 5% (section 3.11.1 3.12.1, Fig. 6 (b)).

With respect to the assumed plume height, the focus of this study is set to the

horizontal transport close to the point source, where spatial gradients (and thus the advection) are largest. Thus we consider a plume height representative for power plant plumes shortly after release rather than a completely mixed PBL.

In response to this comment as well as the comments raised by reviewer #3, we have extended the discussion of the plume height in the revised manuscript in a new dedicated subsection:

*3.2 Effective plume height*

*In this study, horizontal transport is described by horizontal wind fields at a fixed "plume height". This is a simplifying assumption, as the emissions take place at stack height of about 200 m, but are uplifted and vertically mixed within the boundary layer during downwind transport.*

*For the quantification of point source emissions, the focus of this study is set to the horizontal transport close to the point source, where spatial gradients are largest. As shown in Kuhn et al., 2022, power plant emissions at 200 m stack height quickly rise to about 500 m within the first hundred meters.*

*Brunner et al. (2019) investigated the effective height of CO2 emissions for atmospheric transport simulations. This is closely related to the question which altitude has to be considered in order to describe horizontal transport of a fresh power plant plume appropriately. For summer around noon, they report mean effective heights of about 450 m (with a long tail towards larger values).*

*In this study, we assume an effective plume height of 500 m above ground level (agl). For individual stations and specific meteorological situations, systematic deviations might occur. In order to quantify the impact of this assumption, we thus also performed the analysis for a plume height of 300 m (see section  3.12.3).*

*ERA5 wind fields are vertically interpolated to the assumed plume height. In addition, the AMF correction is applied consistently for the same height (see section  3.3).*

8. Line 421: Appendix ??

We corrected line 421 to
*"Additional tables … are provided in the Supplement for various regions."*

9. Table 1: the last column is key to understanding the factor of 3-4 increase relative to version v1, so I would give it a more helpful title and I would identify the principal contributors to the factor of 3-4 increase in the text.

We thank the reviewer for this proposal and modified the column title to "*impact on v2 emission estimate compared to v1*". In addition, we extended the discussion in section  3.14 by including the numbers of the main contributors to the factor of ~3.

**Additional references**

Augustsson, T. R. R., Effects of Multiple Scattering and Surface Albedo on the Photochemistry of the Troposphere, https://digitalcommons.odu.edu/cgi/viewcontent.cgi?article=1213&context=mae_etds198 1981.

Brunner, D., Kuhlmann, G., Marshall, J., Clément, V., Fuhrer, O., Broquet, G., Löscher, A., and Meijer, Y.: Accounting for the vertical distribution of emissions in atmospheric $CO_2$ simulations, Atmos. Chem. Phys., 19, 4541–4559, https://doi.org/10.5194/acp-19-4541-2019, 2019.

de Gouw, J. A., Parrish, D. D., Brown, S. S., Edwards, P., Gilman, J. B., Graus, M., et al., Hydrocarbon removal in power plant plumes shows nitrogen oxide dependence of hydroxyl radicals. Geophysical Research Letters, 46, 7752– 7760. https://doi.org/10.1029/2019GL083044, 2019.

Dix, B., Francoeur, C., Li, M., Serrano-Calvo, R., Levelt, P. F., Veefkind, J. P., et al. Quantifying NOx emissions from U.S. Oil and gas production regions using TROPOMI NO2. ACS Earth and Space Chemistry, 6(2), 403– 414. https://doi.org/10.1021/acsearthspacechem.1c00387, 2022.

Goldberg, D. L., Lu, Z., Streets, D. G., de Foy, B., Griffin, D., McLinden, C. A., Lamsal, L. N., Krotkov, N. A., and Eskes, H. J.: Enhanced Capabilities of TROPOMI NO2: Estimating NOx from North American Cities and Power Plants, Environ. Sci. Technol., 53, 12594–12601, https://doi.org/10.1021/acs.est.9b04488, 2019.

Goldberg, D. L., Harkey, M., de Foy, B., Judd, L., Johnson, J., Yarwood, G., and Holloway, T.: Evaluating NOx emissions and their effect on O3 production in Texas using TROPOMI NO2 and HCHO, Atmos. Chem. Phys., 22, 10875–10900, https://doi.org/10.5194/acp-22-10875-2022, 2022.

Janssen, L. H. J. M., Van Wakeren, J. H. A., Van Duuren, H., and Elshout, A. J.: A classification of no oxidation rates in power plant plumes based on atmospheric conditions, Atmospheric Environment, 22, 43–53, https://doi.org/10.1016/0004-6981(88)90298-3, 1988.

Kim, Yong H., Hyun S. Kim, and Chul H. Song, Development of a Reactive Plume Model for the Consideration of Power-Plant Plume Photochemistry and Its Applications, Environmental Science & Technology 51, no. 3, 1477–87, https://doi.org/10.1021/acs.est.6b03919, 2017.

Kuhn, L., Kuhn, J., Wagner, T., and Platt, U.: The $NO_2$ camera based on gas correlation spectroscopy, Atmos. Meas. Tech., 15, 1395–1414, https://doi.org/10.5194/amt-15-1395-2022, 2022.

Ryerson, T. B., et al., Emissions lifetimes and ozone formation in power plant plumes, J. Geophys. Res., 103( D17), 22569– 22583, doi:10.1029/98JD01620, 1998.

---

## Author Comment (AC2)

**Reply to reviewer #2**

The original review is included in grey. Text changes in the revised manuscript are indicated in italic font.

General comments:

This study provides an updated version of global NOx point source emission inventory using TROPOMI NO2 data. The key idea of the approach is calculating NOx advection to detect point source, and integrating it in a defined area to obtain emissions. Compared with the previous versions, the main improvement includes AMF correction, new spatial derivative calculation, topographic correction, new spatial integration of emission estimate, and lifetime correction. The manuscript is well written. I have a few specific comments.

We thank the reviewer for his/her positive feedback. Below, we refer to the specific comments one by one.

Specific comments:

1. Line 113-114: What is the criteria used for removing cloudy pixels?

   Clouded pixels are removed by the applied filter of the qa value with the threshold of 0.75, which is recommended in the TROPOMI NO2 ATBD (https://sentinel.esa.int/documents/247904/2476257/sentinel-5p-tropomi-atbd-no2-data-products). The qa value is a combination of quality checks performed in the operational processor, and ranges from 0 (pixel cannot be used at all) to 1 (no issues). One of the considered criteria is the cloud radiance fraction – if it is larger than 0.5, the respective qa value is lowered by a factor of 0.74, such that these pixels are not considered in our study.

   We have extended lines 113-114 accordingly:
   *"… removing cloudy pixels (cloud radiance fractions above 50%) as well as anomalies (like solar eclipses) in the TROPOMI NO$_2$ dataset."*

2. Section 3.2: Could you show more details of AMF correction? This study assume a plume at 500 m above ground; so what is the shape of final NO2 profile used for calculate new AMF?

   The assumed profile shape for the excess column is a delta peak at 500 m agl.
   We have extended the respective section in the revised manuscript as follows:
   *Hence, we apply an AMF scaling factor $c_{AMF} = AMF_{plume}/AMF_{PAL}$, where $AMF_{PAL}$ is the tropospheric AMF applied in the PAL product, and $AMF_{plume}$ is calculated from the AK based on a delta-peak profile at plume height. I.e., $c_{AMF}$ reflects how much higher the plume AMF is compared to the a-priori value.*

3. Line 120: What is "ab initio"?
   "ab initio" is a latin term meaning "from the beginning". In order to avoid confusion, we modified the text to *"Only high latitudes … are skipped directly."*

4. Line 421: "IN Appendix ??". Please correct.
   We corrected line 421 to *"Additional tables … are provided in the Supplement."*

5. Line 616: ": American Meteorological ……".Delete ":".
   We corrected the reference accordingly.

---

## Author Comment (AC3)

**Reply to reviewer #3**

The original review is included in grey. Text changes in the revised manuscript are indicated in italic font. References to section numbers are given for both the discussion version of the manuscript (in strike-through mode) as well as the revised manuscript.

Nice job with this new version of the algorithm. All comments are minor.

We thank the reviewer for the positive assessment of our study. Below, we refer to the specific comments one by one.

One overarching comment is in the lack of literature on the realistic nature of both the effective plume heights and the wind speed used. I am glad you performed a sensitivity analysis in Section 3.11 showing the effects of wind speeds on emission estimates, but I do think some literature supporting this analysis is warranted. I am not requesting additional analyses, but instead a longer discussion, including additional literature, on both any potential biases in the ERA5 wind speed/direction and how realistic a 500-m effective plume height is. For example is there a global bias or any regional variation in a wind bias? How is performance of the wind product near coastlines? Are there any references - using in situ aircraft observations - demonstrating that a 500-m plume height is approximately correct mean value?

We agree that our choice of a 500 m effective plume height needs further motivation and discussion. Thus, we extended the discussion of the plume height in a new subsection 3.2 in the revised manuscript as follows:

*3.2 Effective plume height*

*In this study, horizontal transport is described by horizontal wind fields at a fixed "plume height". This is a simplifying assumption, as the emissions take place at stack height of about 200 m, but are uplifted and vertically mixed within the boundary layer during downwind transport.*
*For the quantification of point source emissions, the focus of this study is set to the horizontal transport close to the point source, where spatial gradients are largest. As shown in Kuhn et al., 2022, power plant emissions at 200 m stack height quickly rise to about 500 m within the first hundred meters.*
*Brunner et al. (2019) investigated the effective height of $CO_2$ emissions for atmospheric transport simulations. This is closely related to the question which altitude has to be considered in order to describe horizontal transport of a fresh power plant plume appropriately. For summer around noon, they report mean effective heights of about 450 m (with a long tail towards larger values).*

*In this study, we assume an effective plume height of 500 m above ground level (agl). For individual stations and specific meteorological situations, systematic deviations might occur. In order to quantify the impact of this assumption, we thus also performed the analysis for a plume height of 300 m (see section  3.12.3).*
*ERA5 wind fields are vertically interpolated to the assumed plume height. In addition, the AMF correction is applied consistently for the same height (see section  3.3)*

Other minor comments:

Line 39. "far higher and thus more realistic" —> "larger and more realistic". The word "higher" could refer to height in the atmosphere.

Done.

Line 63. Mention that the 500-m wind is used, and that a sensitivity analysis is performed with the 300-m wind.

In the revised manuscript, we added this information to the end of section 2.2.

Line 74. I see in Line 154 that you provide a range of NOx/NO2 ratios that were used range of NOx/NO2 ratios that were used, but maybe include here also?

Section 2.3 (line 74) is meant to describe the input data, here the ozone climatology. The actual calculation of the NOx/NO2 ratio is described later in section 3.3 (line 154), thus we provide the range of NOx/NO2 ratios in section 3.3 3.4.

And quickly discuss (in either section) where ratios may be higher and lower?

The calculation of the NOx/NO2 ratio is the same as in v1 of the catalog (Beirle et al., 2021). In Beirle et al., 2021, some additional information is provided, in particular a map showing the spatial distribution of the NOx/NO2 ratio (Fig. 2 therein). In the revised manuscript, we have added this information and the respective reference to Beirle et al., 2021 to section 3.3 3.4.

Line 86. Remove "Basically"

Done

Line 105. In the US, the CAMPD is used most often (https://campd.epa.gov/). I am assuming eGRID and CAMPD are identical. Can you confirm? And if so, can you add a sentence in the text mentioning this?

According to eGRID FAQs (https://www.epa.gov/egrid/frequent-questions-about-egrid#egrid1), "eGRID uses data from the Energy Information Administration (EIA) Forms EIA-860 and EIA-923 and EPA's Clean Air Markets Program Data." We have added this information to the revised manuscript.

Line 121. Are there any references - using in situ aircraft observations - demonstrating that a 500-m plume height is approximately correct mean value? It'd be great for you to check out the literature and see if you can find anything.

We have extended the discussion of the plume height and the reasons for choosing 500 m, see the reply above to the overarching comment.

Line 138. This is great, thank you for adding! But can you add a bit more detail? I'm not fully following how the new AMF is calculated. This section would benefit greatly from an illustrative figure showing a standard a priori profile vs an a priori profile with the "excess" at 500-m.

The assumed profile shape of the excess plume is just a δ-peak at 500 m.
We have extended the respective section in the revised manuscript as follows:

*Hence, we apply an AMF scaling factor $c_{AMF} = AMF_{plume}/AMF_{PAL}$, where $AMF_{PAL}$ is the tropospheric AMF applied in the PAL product, and $AMF_{plume}$ is calculated from the AK based on a delta-peak profile at plume height. I.e., $c_{AMF}$ reflects how much higher the plume AMF is compared to the a-priori value.*

Line 178. When you say "this procedure" are you referring to de Foy and Schauer? I think yes, but can you make this clearer?

We reformulated line 178 to "The main advantage of taking derivatives directly on TROPOMI grid is…"

Line 220. Local maxima during the May 2018 - Nov 2021 average? Or daily average? I can imagine the latter is much harder.

We have added *"… the temporal mean (May 2018 - Nov 2021) advection map"* to line 220.

Line 255. Define "area" source. Does this include mobile/vehicle emissions? Or something else?

Within the candidate classification algorithm, local maxima in the advection map are classified by different criteria. Peaks covering a larger area are labelled as "area source" (lines 247-248); this term just refers to the finding of a spatially extended advection maximum. There might be different reasons causing such broad peaks in the advection map, e.g. cities (vehicle emissions) as well as extended industrialized areas, or multiple power plants within about 10-20 km distance. In the revised manuscript, we have extended the candidate classification description and clarified what is meant by "area source":

*Such broad peaks in the advection map might be caused by cities (vehicle emissions) as well as extended industrialized areas, or multiple interfering point sources within about 10-20 km distance.*

Line 291. The NO2 lifetime is also function of the atmospheric composition (both total NO2 and VOCs) as well. I could imagine two locations at the same latitude and with similar wind speeds having differing NO2 lifetimes based on ambient atmospheric composition (Figure 1; https://www.science.org/doi/10.1126/science.aax6832). It seems like you are not accounting for this in your lifetime derivation. Is that a correct assumption? If so, please state explicitly, and discuss in here or in the Discussion section that this would be an opportunity for further improvement of the methodology.

We thank the reviewer for raising this issue. In fact, we have tried to perform individual lifetime estimates for each point source based on an approach similar to Beirle et al., 2011. However, the associated uncertainties were found to be considerably large, and for several point sources (in particular the weaker ones) the algorithm fails. Thus we made use of the simple dependency of latitude reported in Lange et al. (2022). We address the variability of NOx lifetimes at same latitude, as reported in e.g. Laughner and Cohen, 2019, by assuming an uncertainty of 50% for τ and added this information to section  3.10.2.

In addition, we added a new subsection to the discussion of systematic errors dedicated to the lifetime:

*5.3.6 Lifetime correction*

*The lifetime correction (section  3.10.2) is based on a simple parameterization of $\tau$ as function of latitude. However, the OH concentration depends on several parameters like VOC concentrations, as well as on NOx concentration itself, and Laughner and Cohen (2019) report on systematically different lifetimes for locations at comparable latitude.*
*Thus we assumed a rather large uncertainty of 50% for $\tau$. Still, the lifetime correction might be biased for locations where $\tau$ deviates systematically from the parameterization proposed by Lange et al., 2022. In future studies, uncertainties might be reduced by accounting for the actual lifetime for each individual power plant. However, this will be particularly challenging for the weaker sources.*

Line 417. In the Supplement, I noticed that some of the power plants (the ones in Canada and the US at least) didn't match exactly. Instead of saying "First match only", perhaps say "Potential match"

We agree that the phrase "first match only" might be misleading and modified the footnote to "*GPPD power plants / WCD cities within 15 km. Here, only the first match is listed. In the original catalog, all matches are included. Note that the listed power plant or city does not need to represent the actual dominating NOx source.*"

Line 421. I think you meant to say in the "Supplemental Material"
We corrected line 421 to
"*Additional tables … are provided in the Supplement for various regions.*"

Line 425. Tab 2? I think you meant to say " For five of the point sources listed in Table 2…".

We corrected line 425 accordingly.

Line 458. Seems like more than 41 points are in Figure 13. I could be wrong though. Please double check

Figures 12 & 13 display annual means. As PRTR and eGRID data is considered for the years 2018-2020, there are up to 3 data points shown in Figures 12 & 13 per identified point source. We have clarified the figure captions accordingly.

**Additional references**

Brunner, D., Kuhlmann, G., Marshall, J., Clément, V., Fuhrer, O., Broquet, G., Löscher, A., and Meijer, Y.: Accounting for the vertical distribution of emissions in atmospheric $CO_2$ simulations, Atmos. Chem. Phys., 19, 4541–4559, https://doi.org/10.5194/acp-19-4541-2019, 2019.

Laughner, J.L., Cohen, R.C., Direct observation of changing NOx lifetime in North American cities. Science 366, 723-727. DOI:10.1126/science.aax6832, 2019.

Kuhn, L., Kuhn, J., Wagner, T., and Platt, U.: The $NO_2$ camera based on gas correlation spectroscopy, Atmos. Meas. Tech., 15, 1395–1414, https://doi.org/10.5194/amt-15-1395-2022, 2022.

---

## Author Response (AR2)

Mainz, 26 May 2023

Dear Editor,

many thanks for the positive evaluation of our manuscript.

Concerning the raised minor issues:

1) Please provide a publicly available DOI link to your data in the main text (no any restrictions, easy for everyone to access directly ).

We have published the dataset at World Data Centre for Climate (WDCC) under https://doi.org/10.26050/WDCC/No_xPointEmissionsV2. The data can be accessed by everyone without restriction; however, we are aware that users need to register first at WDCC.

Since we provide an update to a previous version of the catalog, we decided to provide the data for v2 at the same repository as v1 (https://doi.org/10.26050/WDCC/Quant_NOx_TROPOMI). This reference was accepted in the previous ESSD paper at that time (https://essd.copernicus.org/articles/13/2995/2021/). Thus, we just assumed that WDCC is a data repository accepted by ESSD.

We have checked the current criteria that ESSD defines for accepted repositories and found that "**Open access**: the data sets have to be available free of charge and without any barriers.". So we conclude that the mandatory registration process demanded by WDCC is considered as "barrier" (or "restriction"). However, we have no impact on WDCC's publication policy.

So we do not know how to proceed now: the catalog is already published at WDCC with a valid doi. Is there any possibility to proceed with this link for now, as in the previous paper? Please note that the catalog itself, which is a rather small dataset, is also provided in the ESSD Supplement.

If this is not possible at all, we would have to discuss with WDCC how to proceed and have to find an alternative repository.

2) Your reference list includes works "in review". Such works can be cited upon submission if being available to the reviewers. They should not be cited in the final, accepted manuscript, unless published, accepted for publication, or available as preprint with a DOI.

We have updated the publication list accordingly.

Kind regards,

Steffen Beirle